# Two Is Better Than One:
# Aligned Representation Pairs for Anomaly Detection

**Alain Ryser**  *alain.ryser@inf.ethz.ch*
*Department of Computer Science*
*ETH Zurich*

**Thomas M. Sutter**
*Department of Computer Science*
*ETH Zurich*

**Alexander Marx**
*Research Center Trustworthy Data Science and Security of the University Alliance Ruhr*
*Department of Statistics*
*TU Dortmund University*

**Julia E. Vogt**
*Department of Computer Science*
*ETH Zurich*

**Reviewed on OpenReview:** *https://openreview.net/forum?id=BtOzdsnWYc*

## Abstract

Anomaly detection focuses on identifying samples that deviate from the norm. Discovering informative representations of normal samples is crucial to detecting anomalies effectively. Recent self-supervised methods have successfully learned such representations by employing prior knowledge about anomalies to create synthetic outliers during training. However, we often do not know what to expect from unseen data in specialized real-world applications. In this work, we address this limitation with our new approach $CON_2$, which leverages prior knowledge about symmetries in normal samples to observe the data in different contexts. $CON_2$ consists of two parts: Context Contrasting clusters representations according to their context, while Content Alignment encourages the model to capture semantic information by aligning the positions of normal samples across clusters. The resulting representation space allows us to detect anomalies as outliers of the learned context clusters. We demonstrate the benefit of this approach in extensive experiments on specialized medical datasets, outperforming competitive baselines based on self-supervised learning and pretrained models and presenting competitive performance on natural imaging benchmarks.

## 1 Introduction

Reliably detecting anomalies is essential in many safety-critical fields such as healthcare (Schlegl et al., 2017; Ryser et al., 2022), finance (Golmohammadi & Zaiane, 2015), industrial fault detection (Atha & Jahanshahi, 2018; Zhao et al., 2019), or cyber-security (Xin et al., 2018). A common real-world example of anomaly detection is the standard screening scenario, where doctors regularly examine a general population for anomalies that would indicate a health risk. Standard screening datasets predominantly comprise samples from healthy people, as most screened individuals do not exhibit any diseases. Detecting anomalies in this setting is challenging, as anomalies can arise from an arbitrary set of potentially rare diseases or measurement errors. At the same time, we predominantly encounter normal samples from healthy people in the dataset. Anomaly detection methods tackle such problems by learning representations that reflect normality during training and detect anomalies as deviations from the learned normal structure at test time.

One way of characterizing the anomaly detection problem is to view it as a one-class classification (OCC) problem (Schölkopf et al., 2001; Tax & Duin, 2004). The idea of OCC is to estimate a tight decision boundary around all normal representations and to detect anomalies as samples that do not lie inside this boundary (see Figure 1 (a)). However, such a decision boundary requires a dense cluster of normal representations, whereas anomalous representations should lie outside this cluster. This requirement is sometimes called the *concentration assumption* (Ruff et al., 2021). It is generally difficult to formalize an objective that learns representations fulfilling the concentration assumption without observing anomalies during training, as a trivial shortcut is to collapse all representations onto a single point (Ruff et al., 2018). While earlier works have proposed various regularization techniques to circumvent this shortcut (Perera & Patel, 2019; Ghafoori & Leckie, 2020), it has recently become popular to learn a decision boundary more explicitly by carefully designing synthetic anomalies (Oza & Patel, 2018; Sabokrou et al., 2020; Tack et al., 2020; Wang et al., 2023). However, anomalies can be diverse and unexpected, making it difficult to simulate them in real-world settings. A more recent line of work focuses on using or adapting the representations from pretrained models (Reiss et al., 2021; Liznerski et al., 2022; Zhou et al., 2024) instead of learning specific representations for anomaly detection. While these methods are vastly successful on natural imaging benchmark datasets, they may not generalize well to more specialized domains that are underrepresented in the pretraining data, as we will demonstrate in our experiments.

In this work, we present how to learn informative, concentrated representations with $CON_2$[1], a novel objective to learn representations for anomaly detection targeting specialized domains, where anomalies are challenging to simulate, and large datasets for pretraining are difficult to obtain. $CON_2$ contrastively learns two separate, concentrated representation spaces from normal training data by leveraging natural symmetries in normal data. These symmetries help us to create *context augmentations* (examples in Figure 5), allowing us to set samples into distinct contexts. $CON_2$ clusters representations according to their contexts to create *context clusters* while encouraging a symmetrical structure of the space (Figure 1). This approach leads to informative representations that are structured according to the properties of normal data. The structure of anomalous samples is typically different from normal samples, which lets us detect these outliers in the representation space learned by $CON_2$. Our method is particularly valuable for specialized datasets, such as in the medical domain, where anomalies may be difficult to obtain or simulate.

Our main contribution is $CON_2$, a new approach to representation learning for anomaly detection. We further present context augmentations, which allow us to put samples into different contexts by leveraging symmetries observed in the normal training dataset. Additionally, we show how to use the representations learned by $CON_2$ to detect anomalies using two anomaly score functions. The score function $\mathcal{S}_{NND}$ measures sample-anomalousness through the nearest neighbor distance to normal training representations, whereas the more efficient $\mathcal{S}_{LH}$ anomaly score provides a simple likelihood-based alternative. Finally, extensive evaluation on diverse medical-imaging benchmarks demonstrates that learning concentrated representations of normal data with $CON_2$ yields superior anomaly detection performance compared to popular self-supervised and pretrained approaches that depend on assumptions about anomalies or simulated examples.

## 2 Related Work

Learning useful normal representations of high-dimensional data for anomaly detection has recently become a popular line of research. Early works have tackled the problem using hypersphere compression (Ruff et al., 2018). Other popular methods define pretext tasks such as learning reconstruction models (Chen et al., 2017; Zong et al., 2018; You et al., 2019) or predicting data transformations (Golan & El-Yaniv, 2018; Hendrycks et al., 2019b; Bergman & Hoshen, 2019). Although these approaches have had some success in the past, the learned representations are often not informative enough for reliable anomaly detection, as there is typically a discrepancy between the pretext task and learning to characterize normal samples. More recently, progress in self-supervised representation learning led to new methods that learn more expressive normal representations through contrastive learning (Sun et al., 2022; Sehwag et al., 2021), improving upon prior work. Methods such as CSI (Tack et al., 2020) and UniCON (Wang et al., 2023) further refine these representations for anomaly detection by introducing simulated anomalies as negative samples.

---

[1]We provide our code on https://github.com/alain-ryser/CON2.

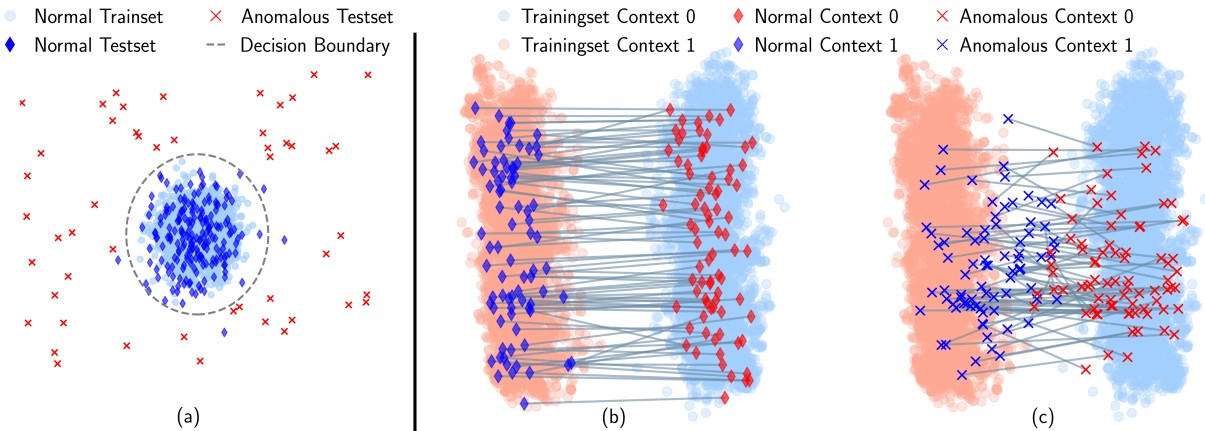

Figure 1: Structure of the representation space in traditional one-class classification (a). Figures (b) and (c) contain two-dimensional PCA embeddings of representations of the train ($\bullet$), normal test ($\blacklozenge$), and anomalous test ($\times$) samples after training $\mathrm{Con}_2$ on normal samples of the BR35H dataset using the invert context augmentation. $\mathrm{Con}_2$ creates two compact, distinct, and aligned context clusters by leveraging symmetries of normal data. Each line corresponds to the position of an original BR35H sample (left) and its context augmented counterpart (right). Parallel lines indicate alignment of the representations of normal test samples, which anomalies fail to achieve.

A separate line of work focuses on estimating the training density with the help of generative models, detecting anomalies as samples from low probability regions (An & Cho, 2015; Schlegl et al., 2019; Nachman & Shih, 2020; Mirzaei et al., 2022). However, these methods tend to generalize better to unseen distributions than to the observed training distribution (Nalisnick et al., 2018), which often harms anomaly detection performance.

Recently, leveraging existing large models pretrained on big, usually unrelated datasets has become a popular approach to tackle anomaly detection. Some methods have been introduced that use representations from such models directly in a zero-shot fashion (Bergman et al., 2020; Liznerski et al., 2022; Jeong et al., 2023), while others demonstrate the benefit of fine-tuning (Cohen & Avidan, 2022; Reiss & Hoshen, 2023; Li et al., 2023; Zhou et al., 2024).

In addition to the traditional setting, where we assume training datasets without any labels, some works started to assume having access to a limited number of labeled samples from the same distribution as the training data. This setting is called anomaly detection with Outlier Exposure (OE) (Hendrycks et al., 2019a), and it has been shown that already a few labeled samples can sometimes greatly boost performance over an unlabeled dataset (Ruff et al., 2020; Qiu et al., 2022; Liznerski et al., 2022). OE has been very successful in the past, often outperforming methods operating in the traditional anomaly detection setting across many benchmarks, though at the cost of requiring labeled samples, which are often not available or hard to obtain in more specialized domains.

Another setting that has recently gained popularity is out-of-distribution (OOD) detection. In OOD detection, we have additional information about our dataset in the form of class labels. Anomaly detection is a special case of OOD detection with only a single label. While the problem is similar, most approaches that tackle OOD detection make specific use of a classifier trained on the dataset labels (Hendrycks & Gimpel, 2017; Lee et al., 2018; Wang et al., 2022), which cannot directly be applied in the anomaly detection setting, as training a classifier on a single class is not straightforward.

In contrast, our method operates in the traditional anomaly detection setting and leverages only the information we have about our normal training samples without making additional assumptions about the nature of anomalies. Further, while we assume access to a dataset containing mostly normal samples, our method does not rely on additional labels, as they can be difficult and expensive to obtain, particularly in more specialized settings.

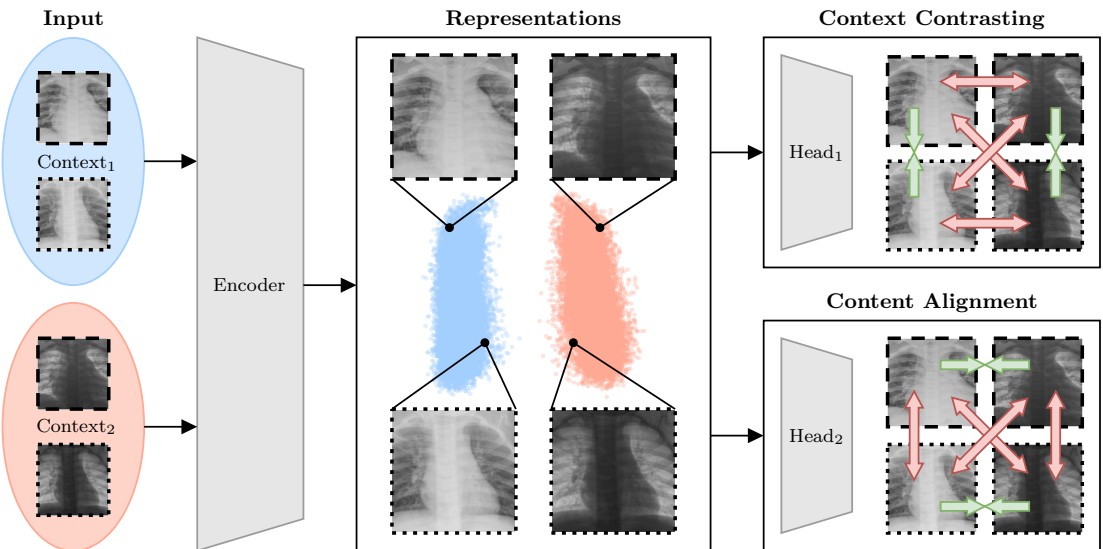

Figure 2: We provide an overview of the training with $\mathrm{Con}_2$. We observe each input sample, dashed/dotted lines marking different samples, in two distinct contexts, and pass it through an encoder to extract its representation. A projection head maps representations into a projection space where we apply *context contrasting* to learn context-specific clusters in the representation space (■ and ■ clusters). Another projection head projects representations to a different space to conduct *content alignment*, encouraging structural alignment between the context clusters. We mark positive and negative pairs with ⇒⇐ and ⟵⟶, respectively.

## 3 Methods

In the following, we recap some background on contrastive learning. We then present our novel representation learning objective $\mathrm{Con}_2$, which allows us to learn tightly clustered, informative representations for anomaly detection when observing samples in two different contexts. Consequently, we introduce the concept of context augmentations, which allow us to create new contexts for arbitrary datasets by leveraging symmetries within normal samples. Finally, we showcase how to use representations learned with $\mathrm{Con}_2$ to detect anomalies at test time.

### 3.1 Contrastive Learning

In this section, we introduce some terminology of contrastive learning (van den Oord et al., 2019), which we later use in our $\mathrm{Con}_2$ objective. Contrastive learning relies on the definition of positive and negative pairs of samples and learns to maximize the similarity between positive representation pairs while pushing apart representations of negative pairs. Popular contrastive approaches, such as SimCLR (Chen et al., 2020), achieve this by incorporating an instance discrimination objective in their loss function. Here, we define the instance discrimination loss as

$$\ell\left(\boldsymbol{x}, \boldsymbol{x}', X\right) := -\log \frac{\exp\left(\mathrm{sim}(\boldsymbol{x}, \boldsymbol{x}')/\tau\right)}{\sum\limits_{\boldsymbol{x}'' \in X, \boldsymbol{x}'' \neq \boldsymbol{x}} \exp\left(\mathrm{sim}(\boldsymbol{x}, \boldsymbol{x}'')/\tau\right)} \ , \tag{1}$$

where we consider $\mathrm{sim}(\boldsymbol{x}, \boldsymbol{x}')$ to be the cosine similarity between two samples $\boldsymbol{x}, \boldsymbol{x}' \in X$ of a dataset $X$. We provide additional background on contrastive learning in Appendix A.1.

### 3.2 Context Contrasting with Content Alignment

The instance discrimination loss from Section 3.1 requires *negative* samples to prevent degenerate solutions. However, we typically do not have access to anomalous negative samples in the anomaly detection setting. Previous work instead relied on simulating synthetic anomalies to circumvent this problem (see Section 2).

However, designing synthetic anomalies is often not feasible, especially in more specialized domains. Our work addresses this limitation with the new $\text{CON}_2$ objective, which leverages additional views or *contexts* of normality instead of creating negative pairs from anomalies. $\text{CON}_2$ learns concentrated representations and circumvents collapse by simultaneously learning two separate but connected normal representations of each sample. We achieve this by combining two contrastive building blocks: *Context Contrasting*, which learns distinct context-dependent clusters of normality, and *Content Alignment*, which ensures that representations capture the semantic information of normality by aligning the positions of context-augmented samples across clusters. This approach allows us to group normal representations in an informative manner and tailor them for anomaly detection in a contrastive way. In Figure 1, we compare the desired structure in the traditional one-class classification setting with the representation space we are learning with $\text{CON}_2$.

First, let's assume that we can observe a dataset $X_{\text{train}}$ from a second perspective, or *context*, retaining the information content of all samples. Let us denote this dataset with $X_{\text{train}}^{\mathcal{C}}$. The idea behind $\text{CON}_2$ is to let our model learn distinct, concentrated representation clusters of both $X_{\text{train}}$ and $X_{\text{train}}^{\mathcal{C}}$. Since we know which sample in $X_{\text{train}}$ corresponds to which sample in $X_{\text{train}}^{\mathcal{C}}$, $\text{CON}_2$ additionally encourages the model to learn a symmetrical structure across these context clusters, effectively aligning the representations between them. In Section 3.3, we will see how to create $X_{\text{train}}^{\mathcal{C}}$ from $X_{\text{train}}$ and assume they are available for the remainder of this section.

Given $X_{\text{train}}$ and $X_{\text{train}}^{\mathcal{C}}$, we define a new dataset and label each sample according to its context as follows:

$$\bar{X}^{\mathcal{C}} := \left\{ (\boldsymbol{x}, 0) \mid \boldsymbol{x} \in X_{\text{train}} \right\} \cup \left\{ (\boldsymbol{x}, 1) \mid \boldsymbol{x} \in X_{\text{train}}^{\mathcal{C}} \right\} \tag{2}$$

Then, let $\mathcal{T}$ be a set of augmentations that model sample invariances similar to Chen et al. (2020). We apply two different augmentations $t_{\boldsymbol{x}}, t'_{\boldsymbol{x}} \sim \mathcal{T}$ for each $(\boldsymbol{x}, y)$ in $\bar{X}^{\mathcal{C}}$ and define

$$\tilde{X}^{\mathcal{C}} := \bigcup_{(\boldsymbol{x}, y) \in \bar{X}^{\mathcal{C}}} \left\{ (t_{\boldsymbol{x}}(\boldsymbol{x}), y), (t'_{\boldsymbol{x}}(\boldsymbol{x}), y) \right\}. \tag{3}$$

Further, for ease of notation, we denote

$$f\left(\tilde{X}^{\mathcal{C}}\right) := \left\{ (f(\boldsymbol{x}), y) \mid (\boldsymbol{x}, y) \in \tilde{X}^{\mathcal{C}} \right\} \tag{4}$$

for any function $f$. Next, we introduce context contrasting and content alignment, the main building blocks of our $\text{CON}_2$ objective.

**Context Contrasting**  As described earlier, we want a model to learn two distinct normal clusters, one for $X_{\text{train}}$ and one for $X_{\text{train}}^{\mathcal{C}}$. We achieve this in a contrastive manner with the *context contrasting* loss $\mathcal{L}_{\text{Context}}(\cdot)$. For a given sample $\boldsymbol{x}$, we derive its representation $g_\theta(\boldsymbol{x})$ using an encoder $g_\theta$. We can then define the context contrasting loss as

$$\mathcal{L}_{\text{Context}}(\tilde{X}^{\mathcal{C}}) := \frac{1}{K} \sum_{\substack{(\boldsymbol{z}, y), (\boldsymbol{z}', y') \in Z_\Phi \\ \boldsymbol{z} \neq \boldsymbol{z}' \wedge y = y'}} \ell(\boldsymbol{z}, \boldsymbol{z}', Z_\Phi), \tag{5}$$

where $K := 4N(2N-1)$ is the normalization constant, $Z_\Phi := h_\phi(g_\theta(\tilde{X}^{\mathcal{C}}))$, $h_\phi$ is a projection head that gets discarded after training similar to Chen et al. (2020), and $\ell$ as in Section 3.1.

Intuitively, context contrasting builds positive and negative sample pairs by matching context labels (see Figure 2). Thus, $\mathcal{L}_{\text{Context}}$ encourages dense context clusters by maximizing the similarity of positive pairs while ensuring *distinctiveness* between clusters by maximizing dissimilarity between negative pairs of sample representations.

**Content Alignment**  While $\mathcal{L}_{\text{Context}}(\cdot)$ allows us to learn context-dependent representation clusters, we also want to leverage our knowledge about sample correspondences between $X_{\text{train}}$ and $X_{\text{train}}^{\mathcal{C}}$ to align the structure between the context clusters. Similar to $\mathcal{L}_{\text{Context}}(\cdot)$, we can accomplish this in a contrastive manner by building positive pairs across clusters, associating all instances that correspond to the same sample,

independent of their context, while negatively associating all pairs of samples that correspond to a different sample. In specific, given a sample $\boldsymbol{x} \in X_{\text{train}}$ and its corresponding $\boldsymbol{x}^{\mathcal{C}} \in X_{\text{train}}^{\mathcal{C}}$ let

$$\Lambda(\boldsymbol{x}) \coloneqq \left\{ f_\Psi(t(\boldsymbol{x})) \mid (t(\boldsymbol{x}), 0) \in \tilde{X}^{\mathcal{C}} \wedge t \in \mathcal{T} \right\} \cup \left\{ f_\Psi(t(\boldsymbol{x}^{\mathcal{C}})) \mid (t(\boldsymbol{x}^{\mathcal{C}}), 1) \in \tilde{X}^{\mathcal{C}} \wedge t \in \mathcal{T} \right\}, \tag{6}$$

where $f_\Psi(\boldsymbol{x}) \coloneqq h_\psi(g_\theta(\boldsymbol{x}))$ and $h_\psi$ denotes a projection head that is independent of $h_\phi$. Intuitively, $\Lambda(\boldsymbol{x})$ contains the 4 projections that are associated with the content augmented samples of $\boldsymbol{x}$ and $\boldsymbol{x}^{\mathcal{C}}$. We then define the *content alignment* loss as

$$\mathcal{L}_{\text{Content}}(\tilde{X}^{\mathcal{C}}) \coloneqq \frac{1}{12N} \sum_{\substack{\boldsymbol{x} \in X \\ \boldsymbol{z}, \boldsymbol{z}' \in \Lambda(\boldsymbol{x}) \\ \boldsymbol{z} \neq \boldsymbol{z}'}} \ell(\boldsymbol{z}, \boldsymbol{z}', Z_\Psi), \tag{7}$$

where $Z_\Phi \coloneqq f_\Psi(\tilde{X}^{\mathcal{C}})$ contains the projections of samples from the augmented dataset, and $\ell$ again as in Section 3.1.

Content alignment ensures that all representations of the same normal sample get matched across different contexts, encouraging *alignment* of the representations between context clusters.

Our final objective $\text{CON}_2$ is then a linear combination of $\mathcal{L}_{\text{Context}}(\cdot)$ and $\mathcal{L}_{\text{Content}}(\cdot)$. This way, $\text{CON}_2$ allows a model to learn *context-specific, content-aligned* representations of normality:

$$\mathcal{L}_{\text{Con}_2}(\tilde{X}^{\mathcal{C}}) \coloneqq \mathcal{L}_{\text{Context}}(\tilde{X}^{\mathcal{C}}) + \alpha \mathcal{L}_{\text{Content}}(\tilde{X}^{\mathcal{C}}) \tag{8}$$

Note that $\mathcal{L}_{\text{Context}}$ and $\mathcal{L}_{\text{Content}}$ contain different numbers of positive and negative pairs. Hence, they have different scales and we thus introduce a weighting factor $\alpha \in \mathbb{R}^+$ (see Appendix C for more details). Figure 2 provides a visual overview of how $\text{CON}_2$ learns representations using context contrasting and content alignment. We provide empirical evidence for the existence of context clusters after training in Appendix E.4.

### 3.3 Context Augmentation

In Section 3.2, we saw how $\text{CON}_2$ applies context contrasting to distinguish between the original training dataset $X_{\text{train}}$ and the dataset in a *distinct* context $X_{\text{train}}^{\mathcal{C}}$. At the same time, content alignment leverages the fact that we can match, or *align*, each original sample with its counterpart in the new context. To create $X_{\text{train}}^{\mathcal{C}}$, we observe that, for most datasets, we can find sample symmetries that allow us to create a distinct new context of its samples without altering their information content. Let $X_{\text{train}} \subset \mathcal{X}$ and $X_{\text{train}}^{\mathcal{C}} = t_{\mathcal{C}}(X_{\text{train}})$, where $\mathcal{X}$ denotes the dataspace and $t_{\mathcal{C}} : \mathcal{X} \to \mathcal{X}$ is a data transformation. We call $t_{\mathcal{C}}$ a *context augmentation* for $X_{\text{train}}$ if it fulfills two heuristic requirements.

**Assumption 1** (*Distinctiveness*). Let $\boldsymbol{x} \sim p_{X_{\text{train}}}$ and $\boldsymbol{x}^{\mathcal{C}} \sim p_{X_{\text{train}}^{\mathcal{C}}}$ be any two samples from the original and the transformed data distribution respectively. The transformation $t_{\mathcal{C}}$ satisfies the *distinctiveness* assumption for $X_{\text{train}}$ if, for any two samples $\boldsymbol{x}$ and $\boldsymbol{x}^{\mathcal{C}}$, it holds that:

$$p_{X_{\text{train}}^{\mathcal{C}}}(\boldsymbol{x}) \approx 0 \text{ and } p_{X_{\text{train}}}(\boldsymbol{x}^{\mathcal{C}}) \approx 0 \tag{9}$$

**Assumption 2** (*Alignment*). Let $\boldsymbol{x}, \boldsymbol{x}' \in X_{\text{train}}$, and let $\mathrm{d}(\boldsymbol{x}, \boldsymbol{x}')$ denote some similarity measure for samples in the input space. Transformation $t_{\mathcal{C}}$ satisfies the *alignment* assumption, if for any two samples $\boldsymbol{x}, \boldsymbol{x}'$, it holds that:

$$\mathrm{d}(\boldsymbol{x}, \boldsymbol{x}') \approx \mathrm{d}(t_{\mathcal{C}}(\boldsymbol{x}), t_{\mathcal{C}}(\boldsymbol{x}')) \tag{10}$$

Intuitively, Assumption 1 ensures a clear distinction between the distributions $p_{X_{\text{train}}}$ and $p_{X_{\text{train}}^{\mathcal{C}}}$, which is necessary to learn separated clusters with context contrasting. Similarly, Assumption 2 requires originally similar normal samples to stay similar in the new context to prevent potential misalignments when applying content alignment. The idea behind distinctiveness and alignment is to help us find a reasonable transformation $t_{\mathcal{C}}$ that is *symmetric* with respect to the normal data distribution $p_{X_{train}}$ in the sense that

$$p_{X_{train}}(\boldsymbol{x}) = p_{X_{train}^{\mathcal{C}}}(t_{\mathcal{C}}(\boldsymbol{x})), \text{ and } p_{X_{train}} \neq p_{X_{train}^{\mathcal{C}}}, \text{ for all } \boldsymbol{x} \sim p_{X_{train}}. \tag{11}$$

Hence, we call $t_\mathcal{C}$ a *context augmentation* if it satisfies both Assumptions 1 and 2 for a given dataset $X_{\text{train}}$. In the following, we introduce some examples of context augmentations that we will use in our experiments in Section 4.

**Invert** This transformation exchanges every pixel value $x$ with the value $1 - x$. Consider a dataset of lung X-rays. Normal tissue and bones cannot lead to the inverse of any normal X-ray image, so distinctiveness is satisfied. Additionally, inversion does not remove semantic information, ensuring alignment.

**Flip** This transformation corresponds to mirroring a sample vertically. Consider a brain MRI dataset. MRIs are typically recorded in standardized world coordinates, such that all samples have the same orientation. Flip changes this orientation, satisfying distinctiveness, while keeping the content of the image unchanged (alignment).

**Equalize** The histogram equalization transformation ensures that the histogram of pixel intensities of an image is uniform. Consider a dataset containing pictures of Melanoma that can be benign or malignant. On such images, histogram equalization changes the color distribution considerably (see Figure 5), ensuring distinctiveness. However, one can still clearly see the same skin features, albeit colored differently, keeping the semantics of the images intact and ensuring alignment.

Among these three, invert satisfies our assumptions for practically all datasets considered in our experiments. We will see representations learned by $\text{Con}_2$ using the invert context augmentation in Section 4.1 and discuss the alternatives in Section 4.3.

### 3.4 Anomaly Detection

Similar to other works in the field (Ruff et al., 2021), we define an anomaly score function $\mathcal{S}$ that maps a given sample's representation onto a scalar to determine its anomalousness and detect anomalies at test time. We can then define a threshold on this anomaly score, predicting *anomaly* for samples above the threshold and *normal* for samples below. See Appendix A.2 for additional background and related work on the anomaly detection setting.

To detect anomalies using the representations of $\text{Con}_2$, we present two anomaly score functions that measure how well a test sample adheres to the context representation clusters. One of the most popular and straightforward approaches to achieve this is a non-parametric nearest neighbor distance approach (Bergman et al., 2020; Sun et al., 2022). Our first score adopts a similar procedure using the cosine similarity, though explicitly leveraging the augmentations used when training $\text{Con}_2$. Specifically, let us define the cosine distance between the training set $X_{\text{train}}$ and a given test sample $\boldsymbol{x}$ with transformation $t$ as

$$s_{\text{NND}}(\boldsymbol{x}; t) := - \max_{\boldsymbol{x}' \in X_{\text{train}}} \frac{\langle g_\theta(t(\boldsymbol{x})), g_\theta(t(\boldsymbol{x}')) \rangle}{\|g_\theta(t(\boldsymbol{x}))\| \|g_\theta(t(\boldsymbol{x}'))\|}. \tag{12}$$

Intuitively, the better a new sample aligns with the context cluster given by augmentation $t$, the more likely it is to be normal. Conversely, a lower cosine similarity indicates that a sample is misaligned with its context cluster, effectively allowing us to flag it as anomalous. While this approach works well in practice, it is rather memory-inefficient, as we need to store the representations of all samples in $X_{\text{train}}$.

To address this limitation, we introduce a likelihood-based score function $s_{\text{LH}}$ to adapt our approach to resource-constrained settings. For simplicity, we assume that representations within each context cluster are distributed according to a multivariate Gaussian distribution. This assumption allows us to efficiently estimate the empirical mean and covariance from the training set and evaluate the probability density to derive an anomaly score without requiring a lot of compute or memory. Note that contrastive approaches typically tend to learn representations with relatively large norms, which may lead to numerical instabilities when estimating the covariance matrix. Our $s_{\text{LH}}$ thus estimates the empirical mean and covariance on the normalized representations. In particular, let

$$Z_{train}^{(t)} := \left\{ \frac{g_\theta(t(\boldsymbol{x}))}{\|g_\theta(t(\boldsymbol{x}))\|} \mid \boldsymbol{x} \in X_{train} \right\} \tag{13}$$

be the normalized representations of the training set augmented with some augmentation $t$. We then compute the density of a multivariate normal distribution based on the empirical mean and covariance,

$$\overline{\mu}_t \coloneqq \overline{\mu}\left(Z_{train}^{(t)}\right) \text{ and } \overline{\Sigma}_t \coloneqq \overline{\Sigma}\left(Z_{train}^{(t)}\right). \tag{14}$$

We then define

$$s_{\text{LH}}(\boldsymbol{x}; t) \coloneqq -\log\left(\mathcal{N}\left(\frac{g_\theta(t(\boldsymbol{x}))}{\|g_\theta(t(\boldsymbol{x}))\|} \,\Big|\, \overline{\mu}_t, \overline{\Sigma}_t\right)\right). \tag{15}$$

We further leverage that our model can differentiate between the two contexts and learns invariances across different augmentations from $\mathcal{T}$ by applying test-time augmentations, similar to previous works (Tack et al., 2020; Wang et al., 2023), which further improves our anomaly detection performance. More specifically, let $\mathcal{T}_{\text{test}} = \{t_1, \ldots, t_A\} \subset \mathcal{T}$ be a set of $A$ test time augmentations. For a given sample $\boldsymbol{x}$ and its corresponding context augmented sample $\boldsymbol{x}^{\mathcal{C}}$, we define our final anomaly score functions $\mathcal{S}_{\{\text{NND,LH}\}} : \mathcal{X} \to \mathbb{R}$ as

$$\mathcal{S}_{\{\text{NND,LH}\}}(\boldsymbol{x}) \coloneqq \frac{1}{A}\left(\sum_{i=1}^{A/2} s_{\{\text{NND,LH}\}}(\boldsymbol{x}; t_i) + \sum_{i=A/2}^{A} s_{\{\text{NND,LH}\}}(\boldsymbol{x}^{\mathcal{C}}; t_i)\right). \tag{16}$$

We will see in our experiments how both scores reliably lead to a competitive anomaly detection performance, though exhibiting a slight performance-efficiency trade-off.

## 4 Experiments

In the following, we compare anomaly detection on representations learned by $\text{CON}_2$ to various anomaly detection approaches based on pretrained foundation models and popular self-supervised methods across various medical imaging datasets, a specialized domain where prior knowledge about anomalies is typically hard to obtain. We further analyze the performance trade-off of $\mathcal{S}_{\text{NND}}$ and $\mathcal{S}_{\text{LH}}$ and explore different context augmentations, discussing their effect on anomaly detection performance. Finally, we examine the impact of context contrasting and content alignment on the performance of $\text{CON}_2$. We refer to Appendices B to D for more details regarding compute, code, the choice of hyperparameters, and our datasets.

**Baselines** We compare our work to various recent contrastive anomaly detection baselines, including SSD (Sehwag et al., 2021), CSI (Tack et al., 2020), and UniCon-HA (Wang et al., 2023). To ensure comparability between $\text{CON}_2$ and other self-supervised methods, we conduct all experiments with a randomly initialized ResNet18 architecture (He et al., 2016). Additionally, we compare our method against CLIP-AD (Liznerski et al., 2022), AnomalyCLIP (Zhou et al., 2024), anomaly detection with $\mathcal{S}_{\text{NND}}$ on I-JEPA (Assran et al., 2023) representations, MVFA (Huang et al., 2024), MediCLIP(Zhang et al., 2024) and PANDA (Reiss et al., 2021), which build on large, pretrained models such as CLIP (Radford et al., 2021) or ResNet (He et al., 2016). Both MediCLIP and MVFA-AD are methods specifically proposed for anomaly detection on medical datasets.

### 4.1 Anomaly Detection with $\text{CON}_2$ Representations

We demonstrate the capabilities of $\text{CON}_2$ across six different medical datasets. We train $\text{CON}_2$ on the healthy population of the training datasets of *BreastMNIST* (Al-Dhabyani et al., 2020) and *OctMNIST* (Kermany et al., 2018b) of the *MedMNIST* collection (Yang et al., 2021; 2023), containing breast ultrasound and retinal optical coherence tomography images, respectively, the *KVASIR* dataset (Pogorelov et al., 2017) which contains endoscopic images of the gastrointestinal tract, the *BR35H* brain MRI dataset (Hamada, 2020), a chest x-ray dataset for *Pneumonia* detection (Kermany et al., 2018a) and a *Melanoma* detection dataset (Javid, 2022). We use the *invert* transformation to put each training dataset into a new context. As discussed in Section 3.3, this transformation satisfies both Assumptions 1 and 2 on most imaging datasets and is thus usually a valid context augmentation. We first perform anomaly detection with $\text{CON}_2$ using the $\mathcal{S}_{\text{NND}}$ anomaly score function and later provide a comparison to $\mathcal{S}_{\text{LH}}$. We run all experiments across three seeds,

Table 1: We apply the $\mathcal{S}_{\mathrm{NND}}$ anomaly score to representations of $\mathrm{Con_2}$ using *invert* as the context augmentation and compare to various baselines that either use large pretrained networks (*Pretrain*) or learn normal representations through self-supervision (*SSL*). We train all methods on normal training samples of six real-world medical imaging datasets and evaluate them on a held-out test set with normal and anomalous samples. Except for the zero-shot baselines, we run each experiment with three different seeds and report the mean ± standard deviation of the *area under the receiver operating characteristic curve* (AUROC).

| Method | | BreastMNIST | OctMNIST | Kvasir | BR35H | Pneumonia | Melanoma |
|---|---|---|---|---|---|---|---|
| Pretrain (Zero-shot) | CLIP-AD | 55.1 | 41.3 | 57.0 | 66.1 | 71.2 | 77.2 |
| | AnomalyCLIP | 63.0 | 68.9 | 68.1 | 96.5 | 70.3 | 62.1 |
| | I-JEPA-ZSAD | 70.8 | 82.3 | 90.3 | 99.9 | 82.1 | 93.5 |
| | MVFA | 55.7 | 91.3 | 74.4 | 78.1 | 45.9 | 72.8 |
| Pretrain (Fine-tuned) | MediCLIP | $59.1_{\pm 6.2}$ | $89.2_{\pm 2.5}$ | $69.4_{\pm 1.5}$ | $88.5_{\pm 6.0}$ | $55.7_{\pm 3.7}$ | $73.5_{\pm 3.1}$ |
| | PANDA | $63.9_{\pm 0.0}$ | $90.3_{\pm 0.0}$ | $91.0_{\pm 0.0}$ | $99.8_{\pm 0.0}$ | $85.9_{\pm 0.0}$ | $93.5_{\pm 0.0}$ |
| SSL | I-JEPA-AD | $70.4_{\pm 0.2}$ | $53.3_{\pm 6.7}$ | $81.6_{\pm 1.7}$ | $99.8_{\pm 0.1}$ | $76.4_{\pm 1.7}$ | $92.1_{\pm 0.3}$ |
| | SimCLR | $74.7_{\pm 3.1}$ | $74.0_{\pm 0.2}$ | $85.2_{\pm 0.5}$ | $99.8_{\pm 0.1}$ | $91.0_{\pm 0.9}$ | $72.9_{\pm 2.8}$ |
| | SSD | $44.6_{\pm 4.3}$ | $82.6_{\pm 0.4}$ | $81.8_{\pm 0.7}$ | $99.8_{\pm 0.1}$ | $90.9_{\pm 0.2}$ | $79.0_{\pm 2.2}$ |
| | CSI | $77.3_{\pm 0.5}$ | $75.0_{\pm 0.1}$ | $88.1_{\pm 0.8}$ | $95.1_{\pm 0.6}$ | $73.9_{\pm 1.6}$ | $92.3_{\pm 0.2}$ |
| | UniCon-HA | $76.2_{\pm 2.3}$ | $68.5_{\pm 0.8}$ | $64.6_{\pm 2.7}$ | $98.6_{\pm 0.0}$ | $86.4_{\pm 0.1}$ | $91.1_{\pm 0.8}$ |
| Ours | $\mathrm{Con_2}$ $(\mathcal{S}_{\mathrm{NND}})$ | $\mathbf{81.7}_{\pm 1.4}$ | $\mathbf{92.3}_{\pm 0.8}$ | $\mathbf{91.4}_{\pm 0.2}$ | $\mathbf{100}_{\pm 0.0}$ | $\mathbf{91.1}_{\pm 0.7}$ | $\mathbf{94.1}_{\pm 0.4}$ |

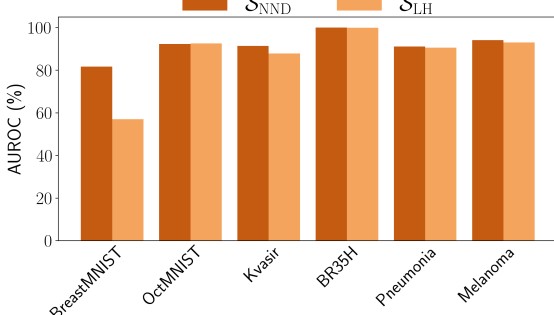

Figure 3: Comparison between anomaly detection with $\mathcal{S}_{\mathrm{NND}}$ and $\mathcal{S}_{\mathrm{LH}}$. The figure shows the AUROC in percentage on all datasets after training $\mathrm{Con_2}$, and evaluating the anomaly scores on the resulting representations.

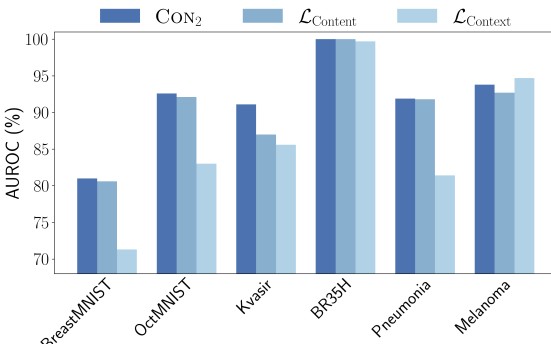

Figure 4: Evaluating the impact of the individual loss terms $\mathcal{L}_{\mathrm{Content}}(\cdot)$ and $\mathcal{L}_{\mathrm{Context}}(\cdot)$ on $\mathrm{Con_2}$. The bars in the figure show the AUROC in percentage on all datasets after training $\mathrm{Con_2}$ and applying $\mathcal{S}_{\mathrm{NND}}$ to the resulting representations.

training on a *healthy train split* and applying our anomaly score functions to the representations of samples of a *held-out test set* to detect anomalies. We report the mean and standard deviation of the resulting area under the receiver operating characteristic curves (AUROC) in Table 1. We report only mean values without standard deviations for our zero-shot baselines, as these methods do not involve randomness.

Anomaly detection using representations learned with $\mathrm{Con_2}$ consistently outperforms our baselines across all datasets. When comparing our method to other self-supervised learning baselines, we see a clear performance gap, demonstrating the advantage of leveraging symmetries that are present in the normal training dataset as opposed to learning representations in a traditional contrastive way, such as SimCLR and SSD, or making assumptions about the expected anomalies, like CSI or UniCon-HA. Further, while CLIP-based zero-shot methods like CLIP-AD or AnomalyCLIP have previously demonstrated impressive performance across many natural imaging datasets, we can see that these methods are not yet able to reach the performance levels of specialized self-supervised approaches. Surprisingly, we found that MVFA and MediCLIP, which are

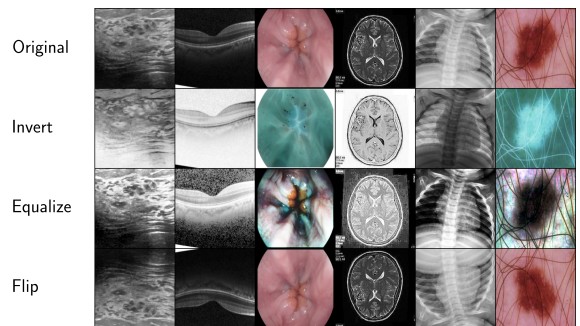

Figure 5: Examples of different context augmentation candidates on *BreastMNIST*, *OctMNIST*, *Kvasir*, *BR35H*, *Pneumonia*, and *Melanoma*, respectively. *Invert* replaces each pixel value $x$ with $1 - x$, *Equalize* stands for histogram equalization, and *Flip* denotes vertical flipping.

| Dataset | Flip | Equalize | Invert |
|---|---|---|---|
| BreastMNIST | $81.7_{\pm 1.4}$ | $72.2_{\pm 1.8}$ | $81.7_{\pm 0.9}$ |
| OctMNIST | $84.3_{\pm 0.5}$ | $87.9_{\pm 0.3}$ | $92.3_{\pm 0.8}$ |
| Kvasir | $87.2_{\pm 2.1}$ | $93.1_{\pm 0.2}$ | $91.4_{\pm 0.2}$ |
| BR35H | $99.8_{\pm 0.3}$ | $99.9_{\pm 0.0}$ | $100_{\pm 0.0}$ |
| Pneumonia | $92.8_{\pm 1.1}$ | $93.9_{\pm 0.3}$ | $91.1_{\pm 0.7}$ |
| Melanoma | $93.4_{\pm 1.1}$ | $94.6_{\pm 0.2}$ | $94.1_{\pm 0.4}$ |

Table 2: Comparison of different context augmentation candidates. We report the mean AUROC on all datasets after training $\text{CON}_2$ across three seeds. Augmentations that satisfy Assumptions 1 and 2 exhibit robust performance.

specifically tailored for anomaly detection on medical datasets, did not consistently outperform broader CLIP-based methods like CLIP-AD and AnomalyCLIP. Further, we found that PANDA, which individually fine-tunes a pretrained ResNet50 on each dataset using a domain adaptation technique, exhibits much better performance, sometimes reaching AUROCs close to what we observe when training with $\text{CON}_2$. Similarly, our I-JEPA-ZSAD baseline, which applies $\mathcal{S}_{\text{NND}}$ on top of representations of a pretrained I-JEPA model, performs surprisingly well. Interestingly, training I-JEPA on our datasets directly (I-JEPA-AD) yields much worse results. Nonetheless, further exploration of anomaly detection using I-JEPA may provide an interesting direction for future work. We provide more details regarding our baselines in Appendix B and additional ablations and experiments in Appendix E.

While anomaly detection with $\mathcal{S}_{\text{NND}}$ on $\text{CON}_2$ representations exhibits impressive performance across all datasets, we need to store the whole dataset to compute the nearest neighbor distance, which is often not feasible for larger datasets. We thus want to compare the performance of $\mathcal{S}_{\text{NND}}$ to the more efficient alternative $\mathcal{S}_{\text{LH}}$ from Section 3.4. When comparing the mean AUROCs (see Figure 3), we can see that $\mathcal{S}_{\text{LH}}$ typically performs very similar to $\mathcal{S}_{\text{NND}}$. We thus suspect that $\text{CON}_2$ typically learns elliptical context clusters, allowing a Gaussian likelihood function to effectively detect anomalies in low probability regions of the representation space. However, for some datasets like BreastMNIST, we observe a significant performance drop, which suggests that elliptical clusters are not always guaranteed. In conclusion, $\mathcal{S}_{\text{NND}}$ exhibits better results overall and should be preferred over $\mathcal{S}_{\text{LH}}$. However, if resource constraints do not permit the usage of $\mathcal{S}_{\text{NND}}$, $\mathcal{S}_{\text{LH}}$ provides an efficient alternative with only minor performance degradation. We compare compute efficiency between $\mathcal{S}_{\text{NND}}$ and $\mathcal{S}_{\text{LH}}$ in Appendix E.3.

## 4.2 Impact of $\mathcal{L}_{\textbf{Content}}$ and $\mathcal{L}_{\textbf{Context}}$

We evaluate the effect of the context contrasting and content alignment on $\text{CON}_2$ by applying $\mathcal{L}_{\text{Content}}(\cdot)$ and $\mathcal{L}_{\text{Context}}(\cdot)$ individually and using the $\mathcal{S}_{\text{NND}}$ score for anomaly detection. As in Section 4.1, we apply the invert context augmentation to all samples in these experiments and present the ablation results in Figure 4.

We observe that $\mathcal{L}_{\text{Content}}(\cdot)$ often performs fairly well. We suspect the structure learned by the content alignment is quite similar to $\text{CON}_2$, though less concentrated and with the context clusters overlapping, which may lead to lower performance. Conversely, $\mathcal{L}_{\text{Context}}$ does not seem to perform well on its own on most datasets. Without content alignment, we suspect that context contrasting collapses the context clusters onto single points similar to the hypersphere collapse in (Ruff et al., 2018). Finally, this experiment demonstrates that combining both terms in $\text{CON}_2$ improves overall anomaly detection performance.

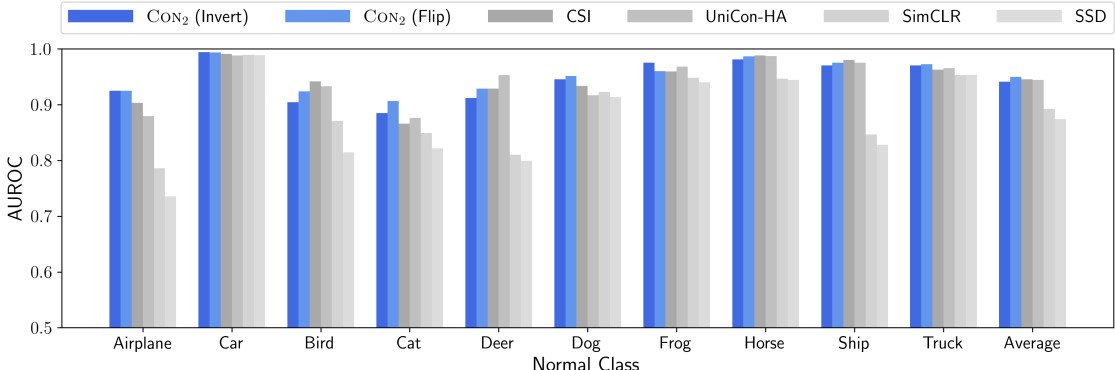

Figure 6: AUROCs of CIFAR10 when setting one class as normal and detecting the rest as anomalous. We compare $\text{CON}_2$ with the invert and flip context augmentations with $\mathcal{S}_{\text{NND}}$ to other contrastive anomaly detection methods. Both the invert and flip context augmentations fulfill our assumptions, resulting in good performances across all classes. Our method further outperforms our baselines in most classes. $\text{CON}_2$ with flip has the highest average across all methods considered.

### 4.3 Alternative Context Augmentations

In our previous experiments, we trained $\text{CON}_2$ representations using the invert context augmentation. However, this transformation may not always satisfy Assumptions 1 and 2 for all datasets. We thus want to explore alternative transformations that could serve as context augmentations in certain scenarios. In particular, we find that vertical flipping and histogram equalization can fulfill our distinctiveness and alignment assumptions on many of our datasets. Figure 5 provides examples of these transformations on each dataset. We compare the performance of all three candidate transformations in Table 2.

While the three transformations typically perform rather similarly across datasets, we observe a clear drop in performance with the flip transformation on BreastMNIST, OctMNIST, and Kvasir, and with the equalize transformation on BreastMNIST and OctMNIST. For the flip transformation, we observe a clear violation of distinctiveness (Assumption 1), as these datasets seem to record samples from an arbitrary angle. On OctMNIST, the equalize transformation seems to introduce some noise artifacts, which can lead to a violation of alignment (Assumption 2). Additionally, some original samples of both BreastMNIST and OctMNIST may look very similar to histogram equalized samples, violating distinctiveness (Assumption 1). Further, note that the flip transformation on Melanoma violates distinctiveness but still performs well, indicating that a violation of Assumptions 1 and 2 does not necessarily imply bad performance. Finally, these experiments demonstrate how proper context augmentations achieve similar performance, validating the definition of our assumptions in Section 3.3.

### 4.4 Natural Imaging

In addition to the results on the more specialized medical imaging domain, our method also exhibits robust performance on more traditional natural imaging benchmark datasets. Here, we train $\text{CON}_2$ on the CIFAR10, CIFAR100 (Krizhevsky et al., 2009), ImageNet30 (Russakovsky et al., 2015; Hendrycks et al., 2019b), Dogs vs. Cats (Cukierski, 2013), and Muffin vs. Chihuahua (Cortinhas, 2023) datasets in the one-class classification setting (Ruff et al., 2021). In the one-class classification setting, we typically work on multi-class classification datasets where we consider one of the classes as the normal class and the rest as anomalies. In particular, we train our model on the training samples of the normal class and want to differentiate between unseen samples of this normal class and all other classes at test time. Similar to our previous experiments, we train each model across three seeds for each class of each dataset. Note that we do not compare pretrained models here, because standard pretraining datasets typically include the samples of these datasets, leading to leakage of the anomaly class during pretraining.

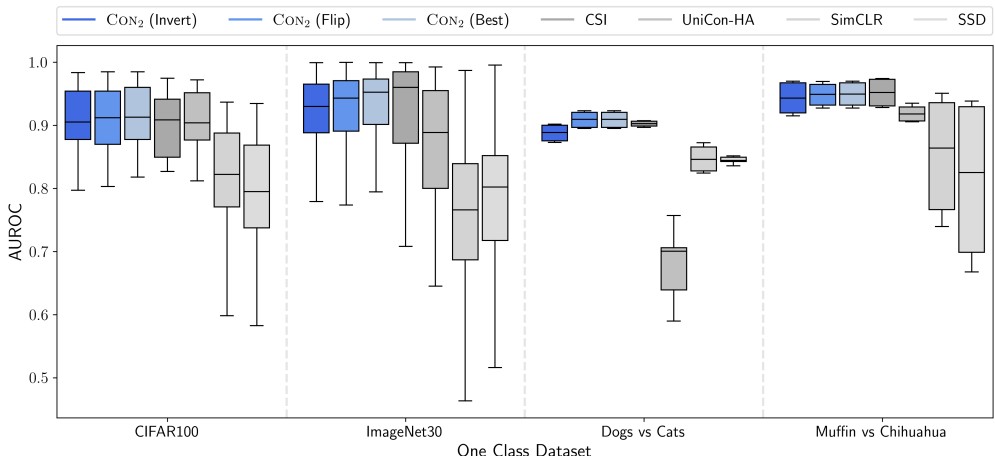

Figure 7: One class classification results for CIFAR100, ImageNet30, Dogs vs. Cats, and Muffin vs. Chihuahua. Our method consistently outperforms our baselines on CIFAR100 and Dogs vs. Cats while exhibiting more robust performance across different normal classes with a similar average performance to CSI on ImageNet30 and Muffin vs. Chihuahua. Additionally, we provide results including $\text{Con}_2$ (Best), which demonstrates how carefully selecting context augmentations satisfying the assumptions of Section 3.3 further improves anomaly detection capabilities of $\text{Con}_2$.

Much like our experiments on medical imaging datasets, we can typically consider the invert transformation as a valid context augmentation in these datasets. Additionally, vertical flipping often satisfies distinctiveness, as natural images are usually not taken from a birds-eye view and adhere to gravity, e.g., a plane of CIFAR10 will typically not fly upside down. Vertical flipping also satisfies alignment since it neither adds nor removes any information from the image, but instead reorders pixel positions. On the other hand, histogram equalization often does not satisfy distinctiveness, as this transformation may result in scenes that seem slightly differently illuminated. We thus only present results of $\text{Con}_2$ with flip and invert context augmentations. We provide examples of each context augmentation in Figure 8 in Appendix E.1.

In Figure 6, we compare the performance of $\text{Con}_2$ and our baselines across the different classes of CIFAR10. For both the invert and the flip context augmentation, $\text{Con}_2$ outperforms our baselines on almost all classes. Here, the flip context augmentation achieves a slightly better average AUROC of 95.3 than the invert transformation, which exhibits an average AUROC of 94.6.

We further provide results on one-class CIFAR100, ImageNet30, Dogs vs. Cats, and Muffin vs. Chihuahua in Figure 7. In addition to the invert and flip context augmentation, we also provide results for $\text{Con}_2$ (Best), which selects the context augmentation individually for each class, depending on which satisfies alignment and distinctiveness better for the current normal class. We report the mean and standard deviation of the AUROCs aggregated over seeds and classes of the respective datasets. Our method compares well against established baselines on natural images, matching or improving the state-of-the-art in self-supervised anomaly detection. Similar to what we saw on CIFAR10, $\text{Con}_2$ displays a robust performance across the board. Our approach outperforms baselines on CIFAR100 and Dogs vs. Cats while matching the performance on ImageNet30 and Muffin vs. Chihuahua, while exhibiting much more consistent performance across different normal classes. We can also see that selecting the context augmentation that best fits our assumptions for each normal class improves the performance. In Appendix E.1, we provide numerical results of $\text{Con}_2$ on all natural imaging datasets and context augmentations, including the histogram equalization context augmentation.

# 5    Conclusion

In this work, we presented the method $\text{Con}_2$, which learns representations suited for anomaly detection by leveraging symmetries in the normal training data. Learning representations without making particular assumptions about anomalous data is particularly useful in specialized domains such as healthcare, where anomalous data can be rare and hard to simulate accurately.

We demonstrated the efficacy of our method on real-world medical imaging datasets, showcasing impressive results when compared to competitive baselines. Our experiments highlighted the applicability of $\text{Con}_2$ in safety-critical applications where robust anomaly detection is essential. We further introduced a likelihood-based alternative to the widely used nearest-neighbor distance anomaly score function. This approach leverages that context clusters tend to be elliptical and usually achieves similar anomaly detection performance to a nearest neighbor distance approach while requiring much less memory. We further demonstrated $\text{Con}_2$'s robustness to the choice of context augmentation, validating the distinctiveness and alignment assumptions of context augmentations. Finally, we showcase how the combination of context contrasting and content alignment with $\text{Con}_2$ leads to the overall improvement of anomaly detection performance.

In conclusion, $\text{Con}_2$ represents a significant advancement in anomaly detection by learning concentrated representations from the normal data without relying on anomalous data. Our approach offers a particularly valuable and effective solution in specialized, high-stakes application domains.

**Limitations**   Our current work focuses exclusively on image-based anomaly detection, and we do not include experiments involving other modalities like time-series or multimodal data, where finding appropriate context augmentations could prove more challenging. However, the definition of $\text{Con}_2$ is broad, and it could be interesting to explore whether the symmetries in time-series, graphs, or multimodal data could naturally serve as context augmentations, though finding appropriate content augmentation may prove difficult for some modalities. Additionally, while we empirically show that $\text{Con}_2$ leads to highly informative representations of normality, we do not provide formal theoretical guarantees for our embeddings. Investigating how our method compares to other representation learning techniques outside of anomaly detection would be an interesting direction for future research. Finally, extending our approach to settings such as outlier exposure or out-of-distribution detection presents another promising direction. These scenarios would further test the robustness and flexibility of $\text{Con}_2$ in handling more complex anomaly detection tasks across a wider range of domains.

**Broader Impact**   While anomaly detection methods offer significant societal benefits, such as supporting doctors in standard screening procedures or identifying adverse samples in safety-critical systems, careful consideration is needed when defining *normal* data. Biases or the underrepresentation of certain groups within these datasets could inadvertently lead to discrimination, especially in sensitive domains like healthcare. Ensuring that normal datasets are representative and unbiased is crucial to avoid unintended harm.

## Acknowledgement

AR is supported by the StimuLoop grant #1-007811-002 and the Vontobel Foundation. TS is supported by the grant #2021-911 of the Strategic Focal Area "Personalized Health and Related Technologies (PHRT)" of the ETH Domain (Swiss Federal Institutes of Technology). AM was funded by ETH Zurich for part of the project.

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

# A    Background

This section provides some terminology for contrastive learning and background about the anomaly detection setting.

## A.1    Contrastive Learning

Recently, contrastive learning has emerged as a popular approach for representation learning (van den Oord et al., 2019; Chen et al., 2020). By design, contrastive learning can learn representations that are agnostic to certain invariances (von Kügelgen et al., 2021; Daunhawer et al., 2023), which makes contrastive learning a particularly interesting choice to learn informative representations of normal samples (Tack et al., 2020; Wang et al., 2023), as it allows us to incorporate prior knowledge about our data into the representing learning process in the form of data augmentations. More specifically, invariances are learned by forming positive and negative pairs over the training dataset by applying data augmentations that should retain the relevant content of a sample.

The goal of contrastive learning is to learn an encoding function $g_\theta(\boldsymbol{x})$, where representations of positive pairs of samples are close and negative pairs are far from each other. For a given pair of samples $\boldsymbol{x}, \boldsymbol{x}' \in X$, we can define the instance discrimination loss as (Sohn, 2016; Wu et al., 2018; van den Oord et al., 2019)

$$\ell(\boldsymbol{x}, \boldsymbol{x}', X) = -\log \frac{\exp(\text{sim}(\boldsymbol{x}, \boldsymbol{x}')/\tau)}{\sum\limits_{\boldsymbol{x}'' \in X: \boldsymbol{x}'' \neq \boldsymbol{x}} \exp(\text{sim}(\boldsymbol{x}, \boldsymbol{x}'')/\tau)} . \tag{17}$$

As mentioned in Section 3.2, we consider the function $\text{sim}(\boldsymbol{x}, \boldsymbol{x}')$ to correspond to the cosine similarity between the two input vectors, as this is one of the most popular choices in the contrastive learning literature.

One of the most prominent contrastive methods is SimCLR (Chen et al., 2020), which creates positive pairs through sample augmentations. There exists a supervised extension called SupCon (Khosla et al., 2020), which incorporates class labels into the SimCLR loss. For a given set of augmentations $T$, a dataset $X = \{(\boldsymbol{x}_i, y_i)\}_{i=1}^{N}$, and an augmented dataset $\tilde{X}$ where $|\tilde{X}| = 2N$ and $(\tilde{\boldsymbol{x}}_{2i}, y_i), (\tilde{\boldsymbol{x}}_{2i+1}, y_i) \in \tilde{X}$ denote two transformations of the same sample using random augmentations from $T$, SimCLR and SupCon introduce the following loss functions:

$$\mathcal{L}_{\text{SimCLR}}(\tilde{X}) = \frac{1}{2N} \sum_{i=1}^{N} \ell(f_\Theta(\tilde{\boldsymbol{x}}_{2i}), f_\Theta(\tilde{\boldsymbol{x}}_{2i+1}), f_\Theta(\tilde{X})) + \frac{1}{2N} \sum_{i=1}^{N} \ell(f_\Theta(\tilde{\boldsymbol{x}}_{2i+1}), f_\Theta(\tilde{\boldsymbol{x}}_{2i}), f_\Theta(\tilde{X})) , \tag{18}$$

$$\mathcal{L}_{\text{SupCon}}(\tilde{X}) = \sum_{(\tilde{\boldsymbol{z}}_i, y_i) \in \tilde{Z}} \frac{1}{N(y_i) - 1} \sum_{\substack{(\tilde{\boldsymbol{z}}_j, y_j) \in \tilde{Z} \\ \tilde{\boldsymbol{z}}_j \neq \tilde{\boldsymbol{z}}_i \wedge y_i = y_j}} \ell(\tilde{\boldsymbol{z}}_i, \tilde{\boldsymbol{z}}_j, \tilde{Z}) . \tag{19}$$

Here, we denote

$$\tilde{Z} \coloneqq \left\{ (f_\Theta(\tilde{\boldsymbol{x}}), y) | (\tilde{\boldsymbol{x}}, y) \in \tilde{X} \right\},$$

Table 3: Average compute hours for the SSL experiments for each dataset and method per run. SimCLR and SSD use the same representations, so we can evaluate both methods in one go and list their compute hours together.

| Method \ Dataset | BreastMNIST | OctMNIST | Kvasir | BR35H | Pneumonia | Melanoma |
|---|---|---|---|---|---|---|
| SimCLR/SSD | 0.6 | 25 | 4 | 2 | 3 | 5 |
| CSI | 2 | 112 | 6 | 4 | 8 | 6 |
| UniCon-HA | 4 | 120 | 8 | 8 | 12 | 18 |
| $\mathrm{Con}_2$ | 1 | 36 | 6 | 3 | 5 | 6 |

where $f_\Theta(\boldsymbol{x}) = h_{\theta'}(g_\theta(\boldsymbol{x}))$, $g_\theta(\boldsymbol{x})$ is a feature extractor, and $h_{\theta'}(\boldsymbol{z})$ is a projection head that is typically only used during training (Chen et al., 2020). Further, we define $f_\Theta(\tilde{X}) = \{f_\Theta(\tilde{\boldsymbol{x}}) \mid (\tilde{\boldsymbol{x}}, y) \in \tilde{X}\}$ and

$$N(y) = |\{(\tilde{\boldsymbol{x}}_i, y_i) \mid (\tilde{\boldsymbol{x}}_i, y_i) \in \tilde{X} \wedge y_i = y\}|$$

is the number of samples in $\tilde{X}$ with label $y$.

## A.2 Anomaly Detection

In the anomaly detection setting, we are given an unlabeled dataset $\{\boldsymbol{x}_1, \ldots, \boldsymbol{x}_n\} = X \subset \mathcal{X}$, while assuming that most samples are normal, i.e., the dataset is practically free of outliers (Ruff et al., 2021). The goal is to learn a model from the given dataset that discriminates between normal and anomalous data at test time.

In this work, we assume the challenging case where our dataset is completely free of anomalies. Hence, we aim to discriminate between the normal class and a completely unobserved set of anomalies at test time. This setting is sometimes called one-class classification or novelty detection.

To achieve this goal, one straightforward approach is to approximate the distribution $p_\mathcal{X}(\boldsymbol{x})$ directly using generative models (An & Cho, 2015; Schlegl et al., 2019). Because we assume normal data to lie in high-density regions of $p_\mathcal{X}$, we can discriminate between normal and anomalous samples by applying a threshold function $p_\mathcal{X}(\boldsymbol{x}) \leq \tau$, where $\tau \in \mathbb{R}$ is an often task-specific threshold (Bishop, 1994). As density-based approaches are often difficult to apply to high-dimensional data directly (Nalisnick et al., 2018), we follow a slightly different line of work.

In this paper, we focus on learning a function $g_\theta : \mathcal{X} \to \mathcal{Z}$ that provides us with representations that capture the normal attributes of samples in the dataset (Sehwag et al., 2021; Tack et al., 2020; Wang et al., 2023), by mapping normal samples close to each other in representation space. On the other hand, anomalies that lack the learned normal structure should be mapped to a different part of the representation space.

Given $g_\theta(\boldsymbol{x})$, a popular approach to detect anomalies is by defining a scoring function $\mathcal{S} : \mathcal{Z} \to \mathbb{R}$ (Breunig et al., 2000; Schölkopf et al., 2001; Tax & Duin, 2004; Liu et al., 2008). The score function maps a representation onto a metric that estimates the anomalousness of a sample. To identify anomalies at test time, we can use $\mathcal{S}$ similarly to the density $p_\mathcal{X}$, i.e., we consider a new sample $\boldsymbol{x}$ to be normal if $\mathcal{S}(g_\theta(\boldsymbol{x})) \leq \tau$, whereas $\mathcal{S}(g_\theta(\boldsymbol{x})) > \tau$ means $\boldsymbol{x}$ is an anomaly.

# B Compute & Code

We run all our experiments on single GPUs on a compute cluster using either an RTX2080Ti, RTX3090, or RTX4090 GPU for training. Each experiment can be run with 4 CPU workers and 16 GB of memory. We provide an overview of the compute for our SSL experiments in Table 3. We omit the runtime for experiments with Pretrain methods, as these usually never run for more than an hour. Our experiments are written using PyTorch (Ansel et al., 2024) with Lightning (Falcon & The PyTorch Lightning team, 2019).

In the following, we list for each method and baseline how we arrive at results and which code we use.

$\text{Con}_2$: We implement $\text{Con}_2$ using PyTorch (Ansel et al., 2024) together with Lightning (Falcon & The PyTorch Lightning team, 2019). To evaluate our method, we use various open-source Python libraries such as NumPy (Harris et al., 2020), scikit-learn (Pedregosa et al., 2011), Pandas (McKinney, 2010; team, 2020), or SciPy (Virtanen et al., 2020). We base the implementation of the instance discrimination loss $\ell$ on the implementation provided in Khosla et al. (2020) (`https://github.com/HobbitLong/SupContrast`).

**SimCLR**: For this baseline, we implement SimCLR (Chen et al., 2020) and compute anomaly scores in a similar fashion as (Sun et al., 2022). For this baseline, we rely on similar packages as $\text{Con}_2$.

**SSD**: We use the same representations as for SimCLR but evaluate by following the procedure outlined in Sehwag et al. (2021).

**CSI**: To run experiments for CSI, we used the code provided in `https://github.com/alinlab/CSI`, implementing new dataloaders for the missing datasets.

**UniCon-HA**: We conducted experiments by running code provided by Wang et al. (2023) implementing new dataloaders for the missing datasets. We thank the authors for sharing their code with us.

**CLIP-AD**: We ran CLIP-AD analoguous to the CLIP-AD experiments described by Liznerski et al. (2022), using the following prompts to describe normal images:
*BreastMNIST*:
```
an image of healthy breast tissue
```
*OctMNIST*:
```
an image of healthy retinal tissue
```
*BR35H*:
```
an mri of a healthy brain
```
*Kvasir*:
```
an image of a healthy cecum, pylorus, or z-line
```
*Pneumonia*:
```
an xray image of a normal lung
```
*Melanoma*:
```
a photo of a benign melanoma
```

**AnomalyCLIP**: We ran AnomalyCLIP with the code provided in `https://github.com/zqhang/AnomalyCLIP`, implementing new dataloaders for the missing datasets.

**I-JEPA-ZSAD**: This baseline is using I-JEPA (Assran et al., 2023) for zero-shot anomaly detection. We took the pretrained model provided in `https://huggingface.co/docs/transformers/en/model_doc/ijepa` and performed anomaly detection with $\mathcal{S}_{\text{NND}}$ on the average-pooled embeddings, using the normal training set to build the nearest neighbor index.

**I-JEPA-AD**: This baseline trains I-JEPA (Assran et al., 2023) on the normal training data of our datasets to later perform anomaly detection. We trained the model using the original I-JEPA codebase in `https://github.com/facebookresearch/ijepa` and performed anomaly detection with $\mathcal{S}_{\text{NND}}$ on the average-pooled embeddings after training. We train I-JEPA using the ViT-Tiny configuration to keep parameter counts comparable to our ResNet18 backbone. To make sure the performance gap between I-JEPA-AD and I-JEPA-ZSAD is not due to the smaller architecture, we also provide results for ViT-Base in Appendix E.5.

**MVFA**: We ran MVFA with the code provided in `https://github.com/MediaBrain-SJTU/MVFA-AD`, implementing new dataloaders for the missing datasets.

**MediCLIP**: We ran MediCLIP with the code provided in `https://github.com/cnulab/MediCLIP`, implementing new dataloaders for the missing datasets.

**PANDA**: We ran PANDA with the code provided in `https://github.com/talreiss/PANDA`, implementing new dataloaders for the missing datasets.

## C  Experimental Details

**Setting**   We evaluate our method in the so-called one-class classification setting (Ruff et al., 2021). More specifically, we assume to have access to only the normal (healthy) class during training. At test time, the goal is to detect whether a new sample from a held-out testset stems from the normal class seen during training or whether it seems anomalous, i.e., deviates from the training distribution.

**Metrics**   Typically, there is a high-class imbalance between normal and anomalous samples in the one-class classification setting. Further, setting an appropriate threshold for the anomaly score is often task-dependent. Therefore, a popular approach to evaluating the performance of anomaly detection methods is to use the area under the receiver operator characteristic curve (AUROC) (Ruff et al., 2021). This metric is threshold agnostic and robust to class imbalance.

**Hyperparameters**   We conduct all experiments using a ResNet18 (He et al., 2016) without the last linear layer as the encoder $g_\theta$. Additionally, we set the two projection heads $h_\phi$ and $h_\psi$ to a standard MLP with one hidden layer, analogous to SimCLR (Chen et al., 2020).

Similar to our method, all baselines make use of test-time augmentations. By default, CSI and UniCon-HA use 40 test time augmentations, which we adopt for all baselines. In our experiments, we set the augmentation class $\mathcal{T}$ to the augmentations introduced by Chen et al. (2020). For the context augmentation, we experiment with vertical flips (Flip), inverting the pixels of an image (Invert), i.e., $t_{\text{Invert}}(\boldsymbol{x}_{ij}) = 1 - \boldsymbol{x}_{ij}$, and histogram equalization (Equalize), see Figure 5 for an illustration.

We choose hyperparameters for $\text{CON}_2$ based on their performance on the CIFAR10 dataset and keep them constant across all experiments to ensure we have no exposure to the anomaly class of the medical datasets. We linearly anneal the hyperparameter $\alpha$ in $\mathcal{L}_{\text{CON}_2}$ from 0 to 1 over the course of training to encourage the model to first learn the context-specific cluster structure while gradually aligning representations over the course of training. We optimize our loss using the AdamW optimizer (Loshchilov & Hutter, 2019) with $\beta_1 = 0.9$, $\beta_2 = 0.999$, weight decay $\lambda = 0.001$, and using a learning rate of $10^{-3}$ with a cosine annealing (Loshchilov & Hutter, 2017) schedule. We run all experiments for 2048 epochs.

## D  Datasets

In the following, we provide details about preprocessing, sources, and licenses of the datasets we use in our experiments.

### BreastMNIST

The BreastMNIST dataset (Al-Dhabyani et al., 2020) is part of the MedMNIST (Yang et al., 2021; 2023) collection. It consists of 780 ultrasound images of breast tissues, which are labeled for breast cancer with *Malignant* and *Benign/Normal* labels. We first resize images to 256 and apply center-cropping to feed $224 \times 224$ images to our model. We ran all our experiments on BreastMNIST with a batch size of 64. The dataset is part of the `medmnist` package, which can be installed with `pip` and is published under the *CC BY 4.0* license.

### OctMNIST

The OctMNIST dataset (Kermany et al., 2018b) is part of the MedMNIST (Yang et al., 2021; 2023) collection and consists of 109′309 optical coherence tomography images, which are labeled for blinding diseases with either the *Normal* label or any of *Choroidal Neovascularization*, *Diabetic Macular Edema*, or *Drusen* as anomalies. We first resize images to 256 and apply center-cropping to feed $224 \times 224$ images to our model. We ran all our experiments on OctMNIST with a batch size of 128. The dataset is part of the `medmnist` package, which can be installed with `pip` and is published under the *CC BY 4.0* license.

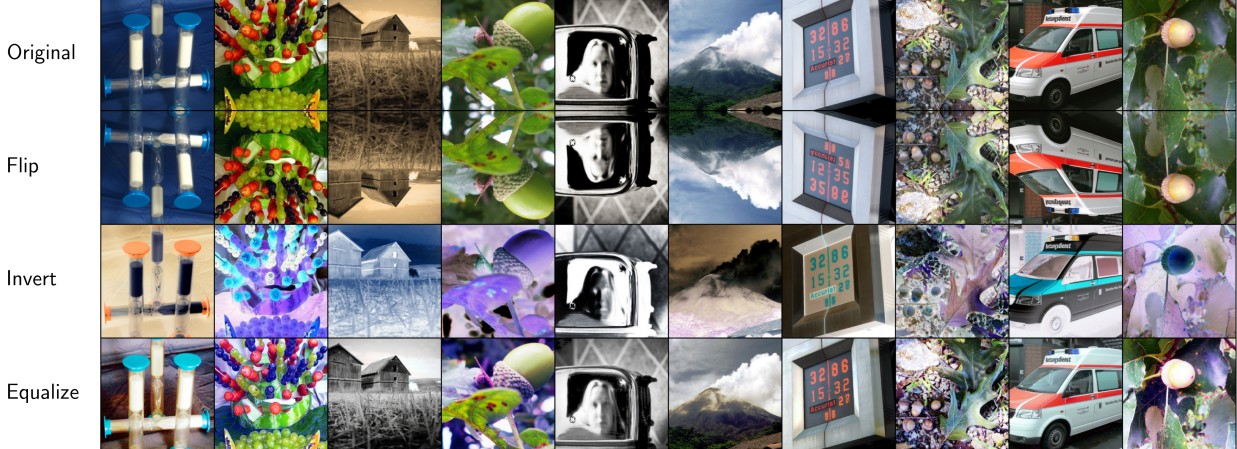

Figure 8: Examples of different transformations that can serve as context augmentations on ImageNet30.

**Kvasir**

The Kvasir dataset (Pogorelov et al., 2017) consists of 4000 endoscopic images of the gastrointestinal tract, which are labeled for various abnormalities with the labels *Normal Cecum*, *Normal Pylorus*, and *Normal z-line* for normal images and any of *Polyps*, *Dyed Lifted Polyps*, *Dyed Resection Margins*, *Esophagitis*, or *Ulcerative-Colitis* for anomalies. We resized images to $224 \times 224$ and ran all our experiments on Kvasir with a batch size of 128. The dataset can be downloaded from `https://www.kaggle.com/datasets/meetnagadia/kvasir-dataset` and is published under the *Open Database* license.

**BR35H**

The BR35H dataset (Hamada, 2020) consists of 3865 brain MRI images and is labeled for brain tumors with binary labels. We first resized images to 256 and applied center-cropping to feed $224 \times 224$ images to our model. We ran all our experiments on BR35H with a batch size of 128. The dataset can be downloaded from `https://www.kaggle.com/datasets/ahmedhamada0/brain-tumor-detection` and is published under the *CC BY 4.0* license.

**Pneumonia**

The Pneumonia dataset was originally published by Kermany et al. (2018a) and consists of $5'863$ lung X-rays, which are labeled with *Pneumonia* and *Normal* labels. We first resize images to 256 and apply center-cropping to feed $224 \times 224$ images to our model. We ran all our experiments on the Pneumonia dataset with a batch size of 128. The dataset can be downloaded from `https://www.kaggle.com/datasets/paultimothymooney/chest-xray-pneumonia` and is published under *CC BY 4.0* license.

**Melanoma**

We use the Melanoma dataset of Javid (2022), which consists of $10'600$ images of Melanoma labeled with being *benign* or *malignant*. We resize all images to $128 \times 128$ before passing them to the model with a batch size 128. The dataset is publicly available at `https://www.kaggle.com/datasets/hasnainjaved/melanoma-skin-cancer-dataset-of-10000-images` and is published under the *CC0: Public Domain* license.

**CIFAR10/CIFAR100**

CIFAR10 and CIFAR100 are natural image datasets with $32 \times 32$ samples. Both datasets consist of $60'000$ samples, totaling 10 and 100 classes for CIFAR10 and CIFAR100, respectively. As CIFAR100 comes with only 600 samples per class, the dataset authors additionally define a set of 20 superclasses, aggregating 5

Table 4: One class classification results for CIFAR100, ImageNet30, Dogs vs. Cats, and Muffin vs. Chihuahua. With all three context augmentations and both scores.

| Method | Score | CIFAR10 | CIFAR100 | ImageNet30 | Dogs vs. Cats | Muffin vs. Chihuahua |
|---|---|---|---|---|---|---|
| $\text{Con}_2$ (Equalize) | $\mathcal{S}_{\text{LH}}$ | $91.1_{\pm 5.8}$ | $86.1_{\pm 5.5}$ | $85.2_{\pm 12.6}$ | $77.0_{\pm 1.1}$ | $83.0_{\pm 12.2}$ |
| | $\mathcal{S}_{\text{NND}}$ | $91.5_{\pm 5.6}$ | $87.5_{\pm 4.4}$ | $86.0_{\pm 12.0}$ | $81.2_{\pm 1.9}$ | $87.5_{\pm 8.0}$ |
| $\text{Con}_2$ (Invert) | $\mathcal{S}_{\text{LH}}$ | $93.7_{\pm 4.3}$ | $89.5_{\pm 5.4}$ | $90.9_{\pm 8.8}$ | $87.8_{\pm 1.0}$ | $91.4_{\pm 4.2}$ |
| | $\mathcal{S}_{\text{NND}}$ | $94.6_{\pm 3.6}$ | $\mathbf{90.6}_{\pm 4.9}$ | $\mathbf{91.2}_{\pm 8.4}$ | $88.7_{\pm 1.5}$ | $93.8_{\pm 3.0}$ |
| $\text{Con}_2$ (Flip) | $\mathcal{S}_{\text{LH}}$ | $94.7_{\pm 3.5}$ | $89.1_{\pm 4.6}$ | $88.9_{\pm 11.9}$ | $90.0_{\pm 1.1}$ | $92.6_{\pm 2.9}$ |
| | $\mathcal{S}_{\text{NND}}$ | $\mathbf{95.3}_{\pm 2.9}$ | $89.7_{\pm 4.2}$ | $89.8_{\pm 11.1}$ | $\mathbf{90.3}_{\pm 1.7}$ | $\mathbf{94.0}_{\pm 1.7}$ |

labels each. In our one-class classification experiments on CIFAR100, we use the superclasses to ensure a manageable number of runs and sufficient training data. We ran all our experiments on CIFAR10 and CIFAR100 with a batch size of 512. Both datasets were published by Krizhevsky et al. (2009) and can be downloaded from `https://www.cs.toronto.edu/~kriz/cifar.html`. To the best of our knowledge, these datasets come without a license.

**Imagenet30**

The ImageNet30 dataset is a subset of the original ImageNet dataset (Russakovsky et al., 2015). It was created by Hendrycks et al. (2019b) for one-class classification. The dataset consists of $42'000$ natural images, each labeled with one of 30 classes. We preprocess the dataset by resizing the shorter edge to 256 pixels, from which we randomly crop a $224 \times 224$ image patch every time we load an image for training. We ran all our experiments on ImageNet with a batch size of 128. The dataset can be downloaded from `https://github.com/hendrycks/ss-ood`, which comes with the MIT License. Further, while we could not find a license for ImageNet, terms of use are provided on `https://image-net.org/`.

**Dogs vs. Cats**

The Dogs vs. Cats was originally introduced in a Kaggle challenge by Microsoft Research (Cukierski, 2013) and consists of $25'000$ images of cats and dogs. We preprocess the dataset by resizing the shorter edge to 128 pixels and then perform center cropping, feeding the resulting $128 \times 128$ image to our model. We ran all our experiments on Dogs vs. Cats with a batch size of 256. The dataset can be downloaded from `https://www.kaggle.com/competitions/dogs-vs-cats/data`. To the best of our knowledge, there is no official license for the dataset, but the Kaggle page points to the Kaggle Competition rules `https://www.kaggle.com/competitions/dogs-vs-cats/rules` in the license section.

**Chihuahua vs. Muffin**

The Chihuahua vs. Muffin dataset consists of $6'000$ images scraped from Google Images. We preprocess the dataset similar to ImageNet30, resizing the shorter edge of the images to 128 pixels while feeding random $128 \times 128$ sized image crops to the model during training. We ran all our experiments on Chihuahua vs. Muffin with a batch size of 256. The dataset was published by Cortinhas (2023) and can be downloaded from `https://www.kaggle.com/datasets/samuelcortinhas/muffin-vs-chihuahua-image-classification/data`. According to the datasets Kaggle page, the dataset is licensed under *CC0: Public Domain.*

In addition to the preprocessing mentioned above, we normalize each image with a mean and standard deviation of 0.5 after applying the augmentations of $\text{Con}_2$.

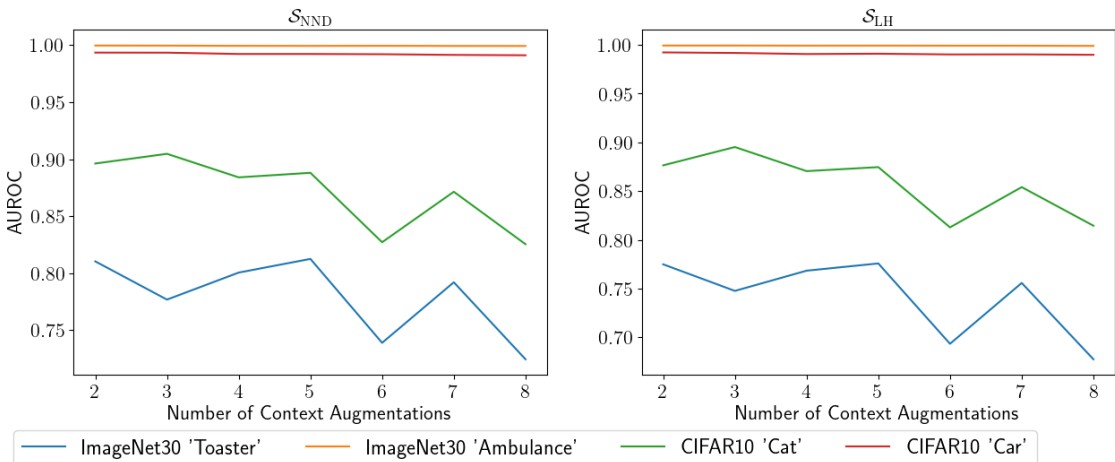

Figure 9: Ablation illustrating the effect of adding more context augmentations. While the performance of well-performing normal classes, such as ImageNet30 *Ambulance* or CIFAR10 *Car*, stays consistent when adding more augmentations, we see a decrease for normal classes such as ImageNet30 *Toaster* or CIFAR10 *Cat* that already perform poor, to begin with.

# E  Ablations

In this section, we present some additional results for the natural image benchmark datasets (Appendix E.1), provide an ablation that explores adding more than one context (Appendix E.2), quantify the efficiency of our anomaly score functions (Appendix E.3), and quantify the presence of context clusters using the silhouette score (Appendix E.4).

## E.1  Natural Image Benchmarks

We provide examples of context augmentations on ImageNet30 in Figure 8. Table 4 shows detailed results of $CON_2$ on all natural imaging benchmarks and anomaly score functions. As we can see, $\mathcal{S}_{NND}$ consistently outperforms $\mathcal{S}_{LH}$. Further, we can see that invert and flip, which usually satisfy distinctiveness and alignment on the natural imaging datasets, outperform the equalize context augmentation, which fails to satisfy our assumptions on many samples, as can be seen in Figure 8.

## E.2  Multiple Context Augmentations

Our formulation in Section 3.3 can be extended beyond only one additional context by slightly adjusting $\mathcal{L}_{Context}$. However, in addition to a loss in efficiency due to requiring more memory, we did not find additional context augmentations to provide a performance benefit, as seen in Figure 9. There, we ran an ablation with different numbers of context augmentations on different classes of CIFAR10 and ImageNet30. In particular, we trained the adapted $CON_2$ loss for 2, 3, 4, 5, 6, 7, and 8 context augmentations, which we derived by combining *Flip*, *Invert*, and *Equalize* from our previous experiments. Adding more augmentations does not seem to harm cases where we experience good performance in the first place. However, we observe a diminishing performance for slightly more challenging classes. Additionally, combining context augmentation may violate distinctiveness or alignment in a pairwise comparison between the different contexts, potentially leading to an unintended structure in the representation space.

## E.3  Anomaly Score Efficiency

Assume representations $Z \in \mathbb{R}^{n \times d}$ of our normal training dataset, where $n$ is the number of samples and $d$ the dimension of the representation space. $\mathcal{S}_{NND}$ requires us to store all samples to perform nearest neighbor search, resulting in a memory complexity of $O(nd)$. On the other hand, $\mathcal{S}_{LH}$ only needs to store

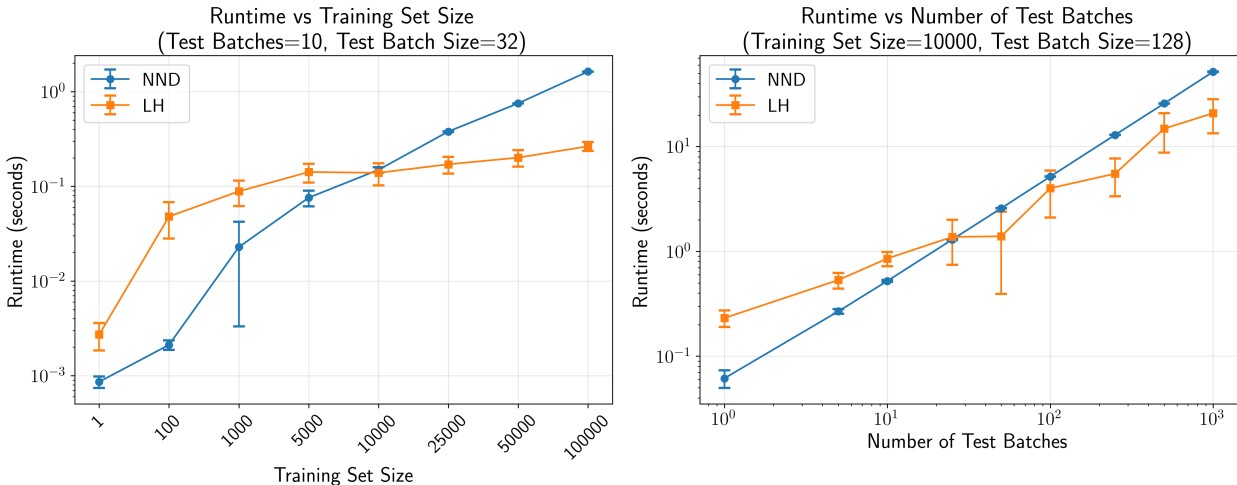

Figure 10: **Left:** $\mathcal{S}_{\mathrm{NND}}$ and $\mathcal{S}_{\mathrm{LH}}$ when scaling $n$ between 1 and 100000, while evaluating 10 batches of 32 samples each. **Right:** Evaluating 1 to 1000 batches of 128 samples each when keeping $n = 10000$.

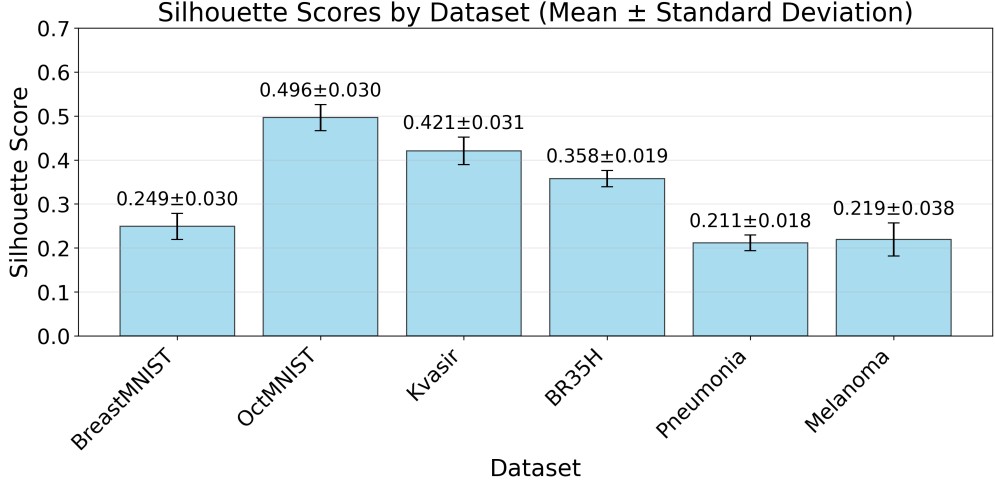

Figure 11: The silhouette score averaged over different seeds for each dataset. The score is $> 0$ for all datasets, indicating a clear presence of context clusters after training with $\mathrm{CON}_2$.

the parameters of a multivariate Gaussian in the representation space, which results in $O(d^2)$ due to the covariance matrix. Hence, $\mathcal{S}_{\mathrm{LH}}$ is more memory efficient at scale than $\mathcal{S}_{\mathrm{NND}}$, as typically $n >> d$. As for runtime, a naive implementation of $\mathcal{S}_{\mathrm{NND}}(\boldsymbol{x})$ would result in a runtime of $O(nd)$, as we would have to compare $\boldsymbol{x}$ to each sample of the training set, while $\mathcal{S}_{\mathrm{LH}}$ is again $O(d^2)$ due to the matrix multiplication when computing the log probability of a gaussian. However, clever implementations in today's compute framework can narrow this gap, and we want to provide additional empirical evidence that compares runtimes between $\mathcal{S}_{\mathrm{NND}}$ and $\mathcal{S}_{\mathrm{LH}}$. In Figure 10, we provide two figures that compare the runtime between $\mathcal{S}_{\mathrm{NND}}$ and $\mathcal{S}_{\mathrm{LH}}$ when varying $n$ and the number of evaluated batches, respectively. As indicated by the asymptotic runtimes provided before, we see that $\mathcal{S}_{\mathrm{NND}}$ is faster for smaller $n$, due to the $d^2$ within $\mathcal{S}_{\mathrm{LH}}$. However, when increasing the number of samples, both when fitting and for evaluating the scores, $\mathcal{S}_{\mathrm{LH}}$ soon becomes much more efficient than $\mathcal{S}_{\mathrm{NND}}$.

Table 5: Comparison of I-JEPA-AD when training on ViT Tiny versus ViT Base. We find that scaling up the number of parameters does not consistently lead to improved performance, and I-JEPA-AD clearly falls short of the performance of I-JEPA-ZSAD.

| Method | BreastMNIST | OctMNIST | Kvasir | BR35H | Pneumonia | Melanoma |
|---|---|---|---|---|---|---|
| I-JEPA-ZSAD (ViT-H/14) | 70.8 | 82.3 | 90.3 | 99.9 | 82.1 | 93.5 |
| I-JEPA-AD (ViT-T) | $70.4_{\pm 0.2}$ | $53.3_{\pm 6.7}$ | $81.6_{\pm 1.7}$ | $99.8_{\pm 0.1}$ | $76.4_{\pm 1.7}$ | $92.1_{\pm 0.3}$ |
| I-JEPA-AD (ViT-B) | $55.6_{\pm 12.7}$ | $38.5_{\pm 15.1}$ | $81.3_{\pm 4.7}$ | $71.1_{\pm 5.8}$ | $71.3_{\pm 12.5}$ | $92.7_{\pm 0.2}$ |

### E.4 Context Clusters

This section provides an ablation that analyzes whether the learned representation space exhibits the context clusters, as claimed in Section 3.2. For each dataset from Section 4.1, we compute the representation of each sample for both contexts. We then label each representation with its corresponding context. This labeling allows us to evaluate how well our representations are clustered by context by calculating the silhouette score (Rousseeuw, 1987). This score function evaluates how well a given cluster assignment relates to the geometry of the dataset by computing a normalized fraction between the intra- and inter-cluster distances. The silhouette score takes values in $[-1, 1]$, where a score $< 0$ indicates wrong labels, ~ 0 indicates overlapping clusters, and $> 0$ indicates a clustering structure with correctly associated labels. Results of this ablation are in Figure 11. We can see that all values are consistently well above 0, clearly indicating a clustering structure within our representation space.

### E.5 I-JEPA-AD Backbone

In Table 5, we compare the results between I-JEPA-AD with ViT-Tiny (5.5 million parameters) and ViT-Base (85.7 million parameters). As can be seen from the large standard deviations, scaling the number of parameters generally leads to inconsistent results and mostly performs worse than training with a smaller backbone. We suspect this may be due to the small number of training samples in our datasets (see Appendix D). However, the impressive performance of I-JEPA-ZSAD suggests that tailoring the I-JEPA training paradigm more specifically for anomaly detection could be an interesting direction for future research in self-supervised anomaly detection.

