# OpenReview forum: "Two Is Better Than One: Aligned Representation Pairs for Anomaly Detection"
_TMLR — Accepted by TMLR_

### Review · Reviewer_ECNy · 2025-06-30

**Summary Of Contributions:**

This work presents a new objective function for self-supervised anomaly detection. The first contribution is the use of dual contrastive losses aimed at enforcing invariance to sample transformations and context transformations. The former enforces content alignment within the sample being invariant to the context changes, while the latter ensures distinct distributions given different contexts. The contexts themselves are created through heuristically derived transformations. The work also presents two scoring functions to determine threshold for anomalous results. The method is then evaluated on medical datasets, with some general benchmark datasets reported in the supplementary. The proposed method outperforms compared methods by some good margin in places, and some ablations to support the decision making is provided.

**Audience:**

Yes

**Broader Impact Concerns:**

I would advise the authors to add a broader impact statement, not only addressing the impact of the work, but also to reflect on the limitations. While I have no major concerns, it would be beneficial for the authors to reflect on this.

**Claims And Evidence:**

No

**Requested Changes:**

**Major**
- From the weakness identified previously about symmetries, either (i) remove the reference to symmetries, or (ii) provide evidence, empirically or theoretically that supports these claims.
- Compare against other existing scoring functions, rather than just comparing your own. This would give better justification for why the proposed method is superior.
- Move the generalised dataset results into the main manuscript, and if possible, give some numerical measure of performance for comparison and benchmarking.
- The authors mention that there is a small efficiency trade-off, but there seems to be no comparative results given that evaluates such efficiency, please add this.
- The abstract could be re-written to introduce the problem statement, and solution better. It does not summarise the problem statement, research question, or the findings.
- Please address the limitations of the work.

**Minor**
- Figure 1 could have an improved caption to assist in readability, for example it is not entirely clear (b), and (c). While it is fine it can be confusing rather than supporting the text,
- There are no equation numbers in the manuscript.
- The last Equation of page 5, where does the denominator '12' appear from?
- The two projection heads should be formally defined in the text, additionally, a more clear and structured introduction to the two tasks, losses, and projection heads would improve readability.
- There are no Standard Deviation given for CLIP-AD, AnomalyClip, and MVFA in Table 1.

**Strengths And Weaknesses:**

**Strengths**:

- The proposed method is simple, yet effective in the evaluation presented. The approach has a wide reaching impact if proven to be generalisable.
- The idea itself is rationale and reasonable in the contrastive framework. It plays off the concepts of invariance to transformations to enforce contextual contrasting, which is a nice use of the framework. I particularly like figure 2, it does a good job of explaining the context alignment and contrasting idea.
- The authors introduce the background and technical concepts well, introducing the topic of contrastive objectives, one-class classification and prior works.
- Ablations studies demonstrate the impact and effectiveness of each component, clearly demonstrating the need of the whole system as presented.

**Weaknesses**:

- The main weakness of the work is that the writing is arguably inconsistent, missing some details, and makes claims that are not substantiated. I do however, feel that this can be simply addressed in revisions.
- The results are limited in scope, while I agree that the medical domain is a suitable use case, the work is presented as a generalised solution. Hence, it would be beneficial to see performance on more generalised datasets, this would mean at the very least moving the results presented in the supplementary into the main manuscript. However, some further analysis would be appreciated.
- Transformations to adjust the context are narrow in scope and do not reflect well real-world (flip, invert, equalise). Additionally, the types of transformation should be introduced more formally within the text, justified, and then analysed to ensure they are doing as stated in the assumption section 3.3. This section introduces the concept, but does not provide any clear implementation details, or justification.
- Following on from the prior, there is little experimentation or analysis on the transformations themselves. This is arguably the most vital factor in the contrastive SSL objective. The choice of what to be invariant too has significant impacts on performance and generalisation to other datasets and domains.
- It is not clear if figure 1 is a real, learned representation space or a visualisation. Some analysis of the clusters/distributions of the representations would be needed to confirm the assumptions of the objective functions. Simply put, is the objective doing what you want it to do?
- The authors make many assumptions and references to symmetries in the representations but no theoretical or empirical evidence is given to support these claims. How do the authors know these symmetries exist and it is not clear what type of symmetries they refer to (from figure 1 I assume some type of translational symmetry).

---

> ### Author Response · Authors · 2025-07-15
> **Response to Weaknesses**
>
> We thank the reviewer for their detailed comments and thoughtful questions. In the following, we address each of their questions and concerns:
>
> > ...writing is arguably inconsistent, missing some details, and makes claims that are not substantiated...
>
> We are happy to address potential inconsistencies to improve the quality of our manuscript. Could you please point us to the specific sections you find problematic?
>
> > The results are limited in scope...moving the results presented in the supplementary into the main manuscript...
>
> We agree that presenting experiments on natural imaging datasets in the main text helps demonstrate the broader applicability of our method. We moved these results to Section 4.4 in the revised manuscript and include numerical results in Appendix E.1. Note that our paper contains extensive experiments on 11 datasets, corresponding to training models on a total of 70 normal classes, and exhibiting consistently high performance across all experiments.
>
> > Transformations should be introduced more formally within the text
>
> We added an introduction of the three context augmentations at the end of section 3.3 of the revised manuscript. Note that we empirically evaluate each augmentation in section 4.3 and discuss failure cases of histogram equalization and flip.
>
> > There is little experimentation or analysis on the transformations themselves.
>
> As noted in the previous point, we evaluated the impact of choosing different context augmentation in the ablation in Section 4.3.
> For content augmentations, we used the standard SimCLR augmentation to stay consistent with prior work in the field. However, we agree that our method could work even better if we adjusted content augmentations to reflect invariances in medical data better.
>
> > Is figure 1 is a real, learned representation space or a visualisation?
>
> Figure 1 contains 2D PCA embeddings after learning representations on normal samples of BR35H using the invert context augmentation. The updated caption now mentions this more explicitly.
>
> > Some analysis of the clusters/distributions of the representations would be needed to confirm the assumptions of the objective functions. Simply put, is the objective doing what you want it to do?
>
> We thank the reviewer for this insightful comment. Evaluating whether our objective function leads to the context clusters as claimed in the methods section is an important aspect. We added an ablation to Appendix E.4 of the revision, where we quantify the presence of context clusters after training by computing the silhouette score [1] over the representations.
>
> > Assumptions and references to symmetries in the representations
>
> By symmetry, we mean the invariance of the normal distribution under the context augmentation. Formally, the context augmentation $t_{\mathcal{C}}$ should be symmetric with respect to the normal data distribution $p_{X_{train}}$ in the sense that $p_{X_{train}}(\mathbf{x})=p_{X_{train}^{\mathcal{C}}}(t_{\mathcal{C}}(\mathbf{x}))$ for all $\mathbf{x}\sim p_{X_{train}}$, while $p_{X_{train}}\neq p_{X_{train}^{\mathcal{C}}}$.
> We designed our contrastive objective to learn a representation space that reflects this property, as illustrated in Figure 1 of our manuscript. We included details of this discussion in section 3.3 of the revised manuscript.

---

> > ### Author Response · Authors · 2025-07-15
> > **Requested Changes**
> >
> > > Evidence that support symmetries.
> >
> > We thank the reviewer for pointing out the lack of details surrounding symmetries. We added more details to section 3.3 to explain what we mean by symmetries. Figure 1 provides qualitative evidence that Con$_2$ reflects this symmetry in the learned representations. We added more details to the caption to make this more explicit.
> >
> > > Compare against other existing scoring functions.
> >
> > As mentioned in our manuscript, $\mathcal{S}_{NND}$ adopts the widely used nearest-neighbor score [2,3], but additionally includes test-time augmentations. Similarly, our baselines, Panda, SimCLR-AD, CSI, and UniCon-HA, also use an NND-based approach. Further, SSD uses a log-likelihood-based scoring function with test-time augmentations that is comparable to our approach.
> >
> > > Move the generalised dataset results into the main manuscript
> >
> > We added results for natural imaging datasets to Section 4.4 in the main text and provide numerical results for these experiments in Appendix E.1.
> >
> > > Efficiency trade-off of anomaly scores
> >
> > We implemented an ablation to compare the compute efficiency between $\mathcal{S}\_{NND}$ and $\mathcal{S}\_{LH}$ and include the results, together with some more details about their asymptotic efficiency, in Appendix E.3 of the revision.
> >
> > > Revision of the Abstract
> >
> > We revised the abstract to summarize our approach better.
> >
> > > Limitations of the work.
> >
> > We added a limitations section in the revision.
> >
> > > Caption of Figure 1
> >
> > We added more details to the caption of Figure 1 to indicate that we are showing real representations.
> >
> > > Equation numbers
> >
> > We added labels to all equations in the revised manuscript.
> >
> > > Denominator 12 in Equation (7)
> >
> > The denominator 12 comes from the fact that $|\Lambda(\mathbf{x})|=4$, and the sum in Equation (7) iterates over distinct pairs in $\Lambda(\mathbf{x})$. Hence, there are
> > $|\Lambda(\mathbf{x})|\cdot(|\Lambda(\mathbf{x})|-1)\cdot N=4\cdot3\cdot N=12\cdot N$
> > many summands. We realize that $|\Lambda(\mathbf{x})|=4$ is not obvious, and we made this more explicit in the revision.
> >
> > > Introduction to the two tasks, and projection heads
> >
> > The revised manuscript contains a high-level introduction to the two tasks at the beginning of Section 3.2 and more details about the architecture of the projection heads in Appendix C.
> >
> > > There are no Standard Deviation given for CLIP-AD, AnomalyClip, and MVFA in Table 1.
> >
> > CLIP-AD, AnomalyClip, and MVFA are all deterministic zero-shot methods without any randomness. As mentioned in the caption of Table 1, we excluded the standard deviation for these methods as it would be $0.0$. We adjusted the table formatting in the revision to make this more obvious.
> >
> > > Broader Impact
> >
> > We added a broader impact section to the revision.
> >
> > ---
> >
> > Thank you once again for your constructive feedback, and please let us know if any further questions or concerns remain.
> >
> > ---
> >
> > [1] Sun et al. *Out-of-distribution detection with deep nearest neighbors.*
> >
> > [2] Rousseeuw, Peter J. *Silhouettes: a graphical aid to the interpretation and validation of cluster analysis.*
> >
> > [3] Bergman et al.*Deep nearest neighbor anomaly detection.*

---

### Review · Reviewer_SuCd · 2025-07-01

**Summary Of Contributions:**

This paper proposes Con2, a self-supervised learning method for anomaly detection that leverages symmetries in normal data to create two distinct but aligned representation spaces. Con2 enforces context-contrasting with content alignment. Hence anomalies which violate the learned symmetrical structures can be easily detected as outliers. Extensive experiments on medical imaging datasets demonstrate that Con2 outperforms existing anomaly detection methods, including self-supervised baselines and pretrained model approaches.

**Audience:**

Yes

**Broader Impact Concerns:**

Sufficiently addressed.

**Claims And Evidence:**

Yes

**Requested Changes:**

- Visualize the learned representation
- Discuss W2 and W4 in paper.

**Strengths And Weaknesses:**

**Strengths**
- The paper is clearly written.
- Approach is interesting and novel.
- Experimental results are strong.

**Weaknesses**
1. I think Figure 2 can be improved by color-coding the 4 input images or add corresponding symbols in Section 3.2. I was initially confused on which pair corresponds to same sample under different context, as it is introduced 2 pages later in Section 3.3 that context augmentation is invert.
2. It is not very clear why "The structure of anomalous samples is typically different from normal samples, which lets us detect them as outliers in the representation...", especially because the context augmentation is color invert, which transforms any image under a fixed rule. It is helpful to elaborate this point with support from empirical evidence.
3. Is Figure 1 using the actual representation learned by Con2? If yes, it is useful to highlight this. If not, I suggest visualizing the representation and checking if it corresponds to Figure 1, especially because leveraging representation symmetries is a big part of the motivation.
4. From the discussion in Section 4.3, it does seem that an appropriate context augmentation is difficult to identify. This can be a limitation, e.g., data from different domain may require non-trivial design of the chosen augmentation.

---

> ### Author Response · Authors · 2025-07-15
>
> We thank the reviewer for highlighting the clarity of our writing, the novelty of our approach, and the strength of our experimental results. In the following, we address each of their concerns point by point:
>
> > I think Figure 2 can be improved...
>
> Thank you for the suggestion. We revised Figure 2 and its caption to clarify which image corresponds to which sample.
>
> > It is not very clear why "The structure of anomalous samples...
>
> A core assumption in anomaly detection is that anomalies reside in low-probability regions of the training distribution and therefore differ intrinsically from normal data. To preserve these high-density regions after augmentation, a context augmentation $t_{\mathcal{C}}$ should be symmetric with respect to the normal data distribution $p_{X_{train}}$ in the sense that $p_{X_{train}}(\mathbf{x})=p_{X_{train}^{\mathcal{C}}}(t_{\mathcal{C}}(\mathbf{x}))$ for all $\mathbf{x}\sim p_{X_{train}}$. We make this more explicit in section 3.3 of the revised manuscript.
>
> > Is Figure 1 using the actual representation learned by Con2?
>
> Yes, Figure 1 shows 2D PCA embeddings of Con$_2$ representations after training on the normal data of BR35H with the *invert* context augmentation. Each line connects the actual positions of an original sample (left) to its context augmented counterpart (right). We added these details to the caption in the revised manuscript.
>
> > Context augmentation are difficult to identify
>
> As with all contrastive learning approaches, the success of Con$_2$ hinges on choosing appropriate augmentations. In general, anomaly detection is an under-specified problem without making any additional assumptions about the data. One advantage of our method is that we can directly inject knowledge about the normal data by specifying appropriate context augmentations. While finding these may not always be trivial, our distinctiveness and alignment assumptions provide clear guidelines for finding augmentations that lead to successful AD. See also our discussion around *How would the method work on other data modalities* with reviewer LXz2 for more details.
>
> ---
>
> ### Requested Changes
> > Visualize the learned representation
>
> We adjusted the caption of Figure 1 to clarify that these are PCA embeddings of BR35H representations learned by Con$_2$.
>
> > Discuss W2 and W4 in paper.
>
> We added details around W2 and W4 to Section 3.3 and the limitations section of the updated manuscript.
>
> ---
>
> We appreciate the reviewer's constructive feedback and are happy to answer any additional questions.

---

### Review · Reviewer_LXz2 · 2025-07-07

**Summary Of Contributions:**

The paper proposes an anomaly detection based on first learning latent representation and then detecting anomalies in latent while assuming that the data in latent follows normal distribution.

**Audience:**

No

**Broader Impact Concerns:**

I do not see any concerns.

**Claims And Evidence:**

No

**Requested Changes:**

* I think it would be fair to show that on the usual benchmark the method does not work all that great.
* I would like to see theoretical grounding of the paper.

**Strengths And Weaknesses:**

The strength of the paper is that the experimental results look very good.

Weaknesses.
* From a distant point of view, this paper is in some sense like many others the authors have cited. The overall theme of these papers are as follows. There exists a thing called self-supervised learning, which promises to learn a good latent representation. We will use it to learn latent representation and detect anomalies in the latent, frequently assuming normal distribution. The self-supervised method usually relies on contrastive learning and differs in how they define augmentations and possibly incorporating other tweaks. This is fine, there are already few papers on this.
* What I see as the weakness is that the absolute lack for formal rigor. While reading the paper, authors offer a rationale why their augmentation and tweak to loss function is better than that of the others, it is more of a speculation.  I mean, it is better to have a rationale rather than nothing (and authors of prior art do the same), but how this is formally related to the density level detection / modelling of probability distribution / or anomaly detection.  Take as a example [1], which explains why learning a classifier between data and their noised version models distribution. Or [2], which tries to explain a particular phenomenon of self-supervised learning. I would like to understand, why the proposed improvements formally leads to better model.
* How the would the method work on other data modalities? Like tabular data or graphs? How can we define the transformations?
* Why the experimental comparison on datasets used by prior art is in the appendix?   Judging by Figure 7, the proposed method is marginally better than the prior art.
* Is the proposed method of self-supervision better than for example JEPA, [4], which follows the same scheme.
* Why the multi-variate gaussian is a good model for the data in the latent space, when they are mapped on the sphere (by dividing the projected point by its norm)?
* Adding pretrained latents, e.g. CLIP is nice, but when that model was pretrained on very different dataset, is it comparable to the method, which was pretraied on the dataset, where one is performing the anomaly detection. Looking at Figure 5, the images looks very different to the normal image used to train CLIP. This comparison is therefore a bit unfair.


[1] Gutmann, Michael, and Aapo Hyvärinen. "Noise-contrastive estimation: A new estimation principle for unnormalized statistical models." Proceedings of the thirteenth international conference on artificial intelligence and statistics. JMLR Workshop and Conference Proceedings, 2010.
[2] Tian, Yuandong, Xinlei Chen, and Surya Ganguli. "Understanding self-supervised learning dynamics without contrastive pairs." International Conference on Machine Learning. PMLR, 2021.
[3] Han, Songqiao, et al. "Adbench: Anomaly detection benchmark." Advances in neural information processing systems 35 (2022): 32142-32159.
[4] Assran, Mahmoud, et al. "Self-supervised learning from images with a joint-embedding predictive architecture." Proceedings of the IEEE/CVF Conference on Computer Vision and Pattern Recognition. 2023.

---

> ### Author Response · Authors · 2025-07-15
>
> We thank the reviewer for pointing out the strength of our experimental results and that our approach is reasonable and well established in the literature. We want to specifically point out the following differences to prior work:
>
> - General methods (AnomalyCLIP, MVFA, Pandas, SimCLR, SSD) make no distributional assumptions, while others (CSI, UniCon-HA) assume properties of anomalies. However, anomaly detection is notoriously underspecified without making assumptions [1], and it may be impossible to make specific assumptions about all types of anomalies. In contrast, Con$_2$ allows us to efficiently incorporate prior knowledge about normality as given by the training set.
> - We can impose a structure (see Fig. 1) on the representation space by observing samples in two different contexts. Our experiments demonstrate how this structure is particularly useful for anomaly detection.
>
> We address each of the reviewers' concerns in the following:
>
> >  Formal rigor
>
> We appreciate the reviewer's interest in gaining more theoretical insights into our method. However, as pointed out by the reviewer, our paper follows a more empirical line of work [1,2,3] that is mostly driven by rationale and experimental results. We are happy to clarify specific theoretical points on request, however, introducing theoretical results in line with the mentioned papers [4,5] goes beyond the scope of our work. We included this point in the limitations section of the revised manuscript.
>
> > How would the method work on other data modalities
>
> The definition of Con$_2$ is modality-agnostic. However, it may not be easy to define context or content augmentations on other modalities in practice. Some ideas for context augmentation that could satisfy distinctiveness and alignment on other modalities include:
>
> * **Tabular:** Permuting feature dimensions or using the PCA representation.
> * **Graphs:** Flipping edge-directions in directed graphs or using the adjacency complement, which would be similar in spirit to the invert operation on images.
> * **Time-series** Leveraging signal processing by, e.g., using the Fourier Transform of a time-series, or leveraging periodic behavior of normal samples if known.
>
> We include part of this discussion in the limitations section of the revision, leaving exploration for future work.
>
> > Why are prior-art datasets only in the appendix?
>
> Experiments on natural imaging datasets are now in Section 4.4 of the revised manuscript and with more details in Appendix E.1.
>
> > Is the proposed method of self-supervision better than, for example, JEPA, which follows the same scheme?
>
> JEPA is not designed for anomaly detection and further differs from Con$_2$ in the following aspects:
>
> - Con$_2$ is a contrastive framework, while JEPA is not.
> - JEPA specifically leverages properties of the ViT architecture through image masking, while Con$_2$ is architecture agnostic.
> - JEPA has two encoders, a trainable context encoder, where context refers to a masked image in this case, and a target encoder, updated using the context encoder's exponential moving average. Con$_2$ is trained with a single encoder.
>
> We agree that exploring other SSL methods, like JEPA, for learning favorable representations for anomaly detection could be an exciting avenue for future research in AD.
>
> > Why is the multi-variate Gaussian a good model...
>
> Our primary anomaly score function $\mathcal{S}\_{NND}$ is a non-parametric nearest-neighbour approach with the cosine-similarity as its distance function. This anomaly score works well and considers the spherical geometry of the normalized representations. $\mathcal{S}\_{LH}$, on the other hand, takes the multi-variate Gaussian as a local Euclidean approximation as an efficient alternative to $\mathcal{S}\_{NND}$. This approach is more compute and memory efficient at scale (see Appendix E.3), and works well in practice, as demonstrated in our experiments.
>
> > Fairness when comparing to CLIP.
>
> We would argue that the comparison to pretrained methods is fair. The point of zero-shot methods like CLIP-AD and AnomalyCLIP is to achieve out-of-domain generalization. AnomalyCLIP, in particular, was also tested on medical data such as CT, MRI, or X-rays in the original paper. MVFA is specifically fine-tuned for zero-shot AD on medical data, and we fine-tune MediCLIP and PANDA on our training data.
>
> ---
>
> We thank the reviewer for their feedback and are happy to answer any additional questions.

---

> > ### Author Response · Authors · 2025-07-15
> > **References**
> >
> > [1] Ruff, Lukas, et al. "A unifying review of deep and shallow anomaly detection." Proceedings of the IEEE 109.5 (2021): 756-795.
> >
> > [2] Tack, Jihoon, et al. "Csi: Novelty detection via contrastive learning on distributionally shifted instances." Advances in neural information processing systems 33 (2020): 11839-11852.
> >
> > [3] Wang, Guodong, et al. "Unilaterally aggregated contrastive learning with hierarchical augmentation for anomaly detection." Proceedings of the IEEE/CVF International Conference on Computer Vision. 2023.
> >
> > [4] Gutmann, Michael, and Aapo Hyvärinen. "Noise-contrastive estimation: A new estimation principle for unnormalized statistical models." Proceedings of the thirteenth international conference on artificial intelligence and statistics. JMLR Workshop and Conference Proceedings, 2010.
> >
> > [5] Tian, Yuandong, Xinlei Chen, and Surya Ganguli. "Understanding self-supervised learning dynamics without contrastive pairs." International Conference on Machine Learning. PMLR, 2021.

---

> > ### Comment · Reviewer_LXz2 · 2025-07-17
> >
> > If it is easy to define augmentations for different modalities, it would make sense to me to demonstrate it.
> >
> > I do not see why JEPA cannot be used for anomaly detection, when it is embedding data into the latent representation, as is this method.

---

> > > ### Author Response · Authors · 2025-07-17
> > >
> > > Thank you for your follow-up questions. We would like to clarify these two points:
> > >
> > > **Augmentations.** As mentioned in our previous response, finding appropriate augmentations for other modalities may be non-trivial. While we listed some ideas for context augmentations on tabular, graph, and time-series data, finding content augmentations analogous to the SimCLR augmentations may be tricky for some modalities. We added this aspect to the limitation section of the revision.
> > >
> > > **I-JEPA for anomaly detection.** In response to your original question, we previously listed the differences between our method and I-JEPA. However, we agree that anomaly detection using I-JEPA is possible by applying our anomaly score function to the embeddings of the model. We added I-JEPA-AD, which uses the $\mathcal{S}\_{\text{NND}}$ score, as an additional baseline in Table 1 of the revised manuscript. To the best of our knowledge, we are the first to explore I-JEPA for anomaly detection. We observe that I-JEPA-AD exhibits promising results, suggesting that future research on specifically adapting I-JEPA for anomaly detection may be an exciting direction for future work.

---

> > > > ### Comment · Reviewer_LXz2 · 2025-07-18
> > > >
> > > > Thanks.

---

> > > > > ### Comment · Reviewer_LXz2 · 2025-07-21
> > > > >
> > > > > I am having a following thought.
> > > > >
> > > > > In my view, the scientific paper should compare rigorously to the prior art. There is not need to adapt a prior art by inventing a new methods, but the comparison should be fair, such that the newly proposed method convincingly demonstrated its pros and cons.
> > > > >
> > > > > With this respect, I have a problem how authors compare to I-JEPA, because they have used pretrained model. I-JEPA is so simple, that there should not be a problem to train it on a different dataset, especially when proposed method is trained on them. This significantly decreases the value of the comparison, because most readers will ask the question, what will happen when I will train I-JEPA on the dataset on which I am performing the anomaly detection.

---

> > > > > > ### Author Response · Authors · 2025-07-21
> > > > > >
> > > > > > Thank you for raising this concern. We believe the comparison is fair for three reasons:
> > > > > >
> > > > > > ---
> > > > > >
> > > > > > **I‑JEPA is not a prior state‑of‑the‑art anomaly‑detection method.**
> > > > > >
> > > > > > As mentioned in our initial response, I‑JEPA is not designed for anomaly detection. It was introduced as a general‑purpose pre‑text task and, to the best of our knowledge, has never been proposed (let alone benchmarked) as an anomaly detector. We therefore treated it exactly as prior work does, namely, as a fixed, off‑the‑shelf encoder whose features are evaluated on downstream tasks. Retraining it for each small anomaly‑detection dataset would change its intended use and depart from accepted practice.
> > > > > >
> > > > > > ---
> > > > > >
> > > > > > **Training I‑JEPA from scratch during the rebuttal period is infeasible.**
> > > > > >
> > > > > > The smallest publicly available model (ViT-H/14) was trained for **72 hours on 16 × A100 GPUs**, i.e., **≈ 1,152 A100‑GPU‑hours** (see Abstract/Figure 1 of the I‑JEPA paper). Larger variants scale up further. Acquiring equivalent compute is impractical for most academic labs, and the engineering effort to reproduce their results on several new datasets cannot be accomplished within the two‑week rebuttal window.
> > > > > >
> > > > > > ---
> > > > > >
> > > > > > **Using the official, publicly released weights provides the fairest and most reproducible baseline.**
> > > > > >
> > > > > > This new baseline shows how well a state‑of‑the‑art foundation encoder, when used exactly as released , performs on anomaly detection compared with our method. This mirrors real‑world usage and provides valuable insights for any reader who wants to use an off-the-shelf method for anomaly detection.
> > > > > >
> > > > > > ---
> > > > > >
> > > > > > As previously mentioned, we think anomaly detection based on I-JEPA may be an interesting direction for future work in the field and we added this more explicitly to the newest revision.

---

### Author Response · Authors · 2025-07-15
**General Response**

We thank all reviewers for their careful reading and helpful suggestions. Specifically, we appreciated the reviewers pointing out

* **Strong empirical performance** as highlighted by **LXz2** (*experimental results look very good*) and **SuCd** (*experimental results are strong*).
* **Clarity of presentation**, as noted by **SuCd** (*paper is clearly written*) and **ECNy** (*background and technical concepts introduced well*).
* **Conceptual novelty / simplicity**, as pointed out by **SuCd** (*approach is interesting and novel*) and **ECNy** (*method is simple, yet effective; idea is rationale and reasonable*).
* **Thorough ablations and illustrative figures**, as mentioned by **ECNy** (*ablations show each component’s effect; Fig. 2 explains the idea*).

Based on the reviewers' feedback, we included the following aspects in the revised manuscript:

1. **Abstract**
    We added some details to the abstract for clarity.
2. **Figures and Tables**
    We provide more details of Figure 1 in the updated caption, and adjusted the formatting of Figure 2 and Table 1 to improve readability.
3. **Section 3.2**
    The section *Context Contrasting with Content Alignment* now includes more intuition behind our approach, provides additional information related to symmetries, and introduces the considered context augmentations in more detail.
4. **Natural Imaging Datasets**
    We added results on natural imaging datasets to the main text in Section 4.4 and provide numerical results in Appendix E.1.
5. **Limitations and Broader Impact**
    We added a limitations and broader impact paragraph.
6. **Model Architecture**
    Appendix C now contains more details regarding the model architecture
7. **Score Efficiency**
    We provide an additional ablation comparing runtime efficiency between $S_{NND}$ and $S_{LH}$ in Appendix E.3
8. **Presence of Context Clusters**
    Appendix E.4 now contains an ablation that helps us quantify the presence of context clusters by computing the silhouette score over the Con$_2$ representations of our datasets.

We hope these changes address all of the reviewers' concerns and further strengthen the contribution of **Con$_2$**.

---

### Decision · Action_Editor_T7N1 · 2025-08-27

**Recommendation:** Accept with minor revision

**Additional Comments:**

The paper proposes Context Contrasting (Con2), a self-supervised representation learning for anomaly detection---no anomaly data is available for training.  The proposed Con2 is trained on samples with different contexts, and the representations are learned so that different contexts are separated and the same contents are concentrated.  In the learned representation space, the anomaly score is defined as the nearest neighbor distance or its Gaussian approximation.  Good performance is demonstrated on medical and general image data.

Reviewers acknowledge the novelty and good experimental results with ablations, while they raised several concerns including clarity of some details, unsupported assumptions, lack of efficiency evaluation, and unfair comparison.  The authors addressed most concerns but the experimental fairness.  Consequently, two out of three reviewers gave "no" for "Claims and Evidence."

Con2 is a novel representation learning method, and I-JEPA can do the same task.  The originality of the paper is in how Con2 learns the representations from normal data, and therefore, the authors should show the proposed way of representation learning is better than any sota representation learning method that can be applicable for anomaly detection, in order to support the claim "outperforming competitive baselines ...'  The authors admitted using I-JEPA trained on the medical data is a promising approach.  IMO, any applicable, promising, existing approach should not be excluded from the list of the baseline methods.

I recommend "conditional acceptance" because this can be addressed relatively easily by doing either of the following:

1. Show Con2 outperforms I-JEPA trained on the same data.
2. Set the goal of the paper, for which I-JEPA trained on the medical data is not applicable.

Option 2 is for the case when the authors really think I-JEPA should be excluded from the list of the baseline methods for the goal the authors have in mind.  This point is not clear from reading the current abstract and introduction, which must be revised.  For example, if training I-JEPA is too costly, the authors could focus on some application scenarios where users cannot (or will not) invest much computation cost for training, even if investing higher costs would result in higher accuracy.  I suppose such application scenarios are probably not in the medical domain, because I cannot image pathologists would dare to use a foundation model trained on general image domain, nor choose a cheap method sacrificing the accuracy for saving computational costs.

**Audience:**

Yes

**Audience Explanation:**

All reviewers acknowledged that the audience criterion is fulfilled.

**Claims And Evidence:**

No

**Claims Explanation:**

The authors' claim "outperforming competitive baselines ...' is not supported without I-JEPA as a baseline.  The authors insisted that I-JEPA trained on the target data should not be regarded as a baseline, but Reviewers LXz2 and ECNy have not been convinced.

I quote Reviewers LXz2's final comment in his recommendation:

"I do not agree with authors that other methods for embedding, like the mentioned I-JEPA are not relevant. I think that comparing the proposed method trained on medical domain to models trained on usual general domains is unfair. The fact that no-one has tried I-JEPA for anomaly detection in my view does not disqualify it from fair comparison."

**Resubmission Of Major Revision:**

The authors may consider submitting a major revision at a later time.